# Human papilloma virus vaccination uptake and associated factors among adolescent girls in Merab Abaya district, Gamo zone, Southern Ethiopia: Mixed methods

Zenebe Debena Den'o[1], Wondimagegn Paulos[2], Desta Markos[2], Woldetsadik Oshine Oche[3], Tamirat Mathewos Milkano[4]*

1 Maternal, Child and Nutrition Department, Merab Abaya District Health Office, Bribir, Ethiopia, 2 School of Public Health, College of Medicine and Health Science, Wolaita Sodo University, Wolaita Sodo, Ethiopia, 3 Department of Nursing, College of Medicine and Health Science, Wachemo University Durame Campus, Durame, Ethiopia, 4 Maternal, Child and Nutrition Department, Bolosso Bombe District Health Office, Bombe, Ethiopia

* temumath54@gmail.com

## Abstract

### Background

Human papillomavirus (HPV) vaccination is a well-established global strategy for the prevention of cervical cancer. However, the uptake of the vaccine varies across regions and countries due to several factors. Although girls are at risk for cervical cancer, there are limited studies measuring vaccination uptake among female adolescents in the study area.

### Objective

To assess human papilloma virus vaccination uptake and associated factors among adolescent girls, in Merab Abaya district, Gamo Zone, southern Ethiopia, 2024.

### Method

A community-based cross-sectional mixed-method study was conducted among 626 adolescent girls selected using a two-stage sampling technique in Merab Abaya District, Gamo Zone, from February 1 to March 30, 2024. For the qualitative component, participants were selected using a purposive sampling technique. Data were entered using EpiData version 4.62 and analyzed using SPSS version 26. Logistic regression was performed to examine the association between the dependent variable and associated factors. Variables with a p-value < 0.05 in the multivariate analysis were considered statistically significant. For qualitative data analysis, OpenCode 4.02 software was used to conduct thematic content analysis.

**Data availability statement:** All relevant data are within the papers and its supporting information files. But the qualitative parts of the study, the participants were not consented to share their audio.

**Funding:** The author(s) received no specific funding for this work.

**Competing interests:** On behalf of all authors, the corresponding author declares that there is no competing interest.

**Abbreviation:** AOR, Adjusted Odds Ratio; CC, Cervical Cancer; CI, Confidence Interval; ETB, Ethiopian Birr; FGD, Focus Group Discussion; IDIs, In-depth Interviews; HPV, Human Papilloma Virus; SD, Standard Deviation; SPSS, Statistical Package for Social Science; WHO, World Health Organization.

## Result

A total of 601 adolescent girls participated in this study, yielding a response rate of 96%. Of these, 324 (53.9%; 95% CI: 49.9–57.9%) had received the human papillomavirus vaccine. Vaccine uptake was significantly associated with: Good knowledge about the HPV vaccine (AOR = 3.4; 95% CI: 2.14–5.38), A positive attitude toward the HPV vaccine (AOR = 1.7; 95% CI: 1.02–2.78), Recommendation from health workers to get vaccinated (AOR = 3.8; 95% CI: 2.25–6.50), Family support for vaccination (AOR = 7.1; 95% CI: 3.97–12.60). Qualitative findings identified mistrust of the HPV vaccine, irregular vaccine provision, and lack of information provision as major barriers to uptake.

## Conclusion

In this study, nearly fifty-four percent of adolescent girls had received the HPV vaccine. The overall uptake of the HPV vaccine among adolescent girls remains low. Good knowledge about the HPV vaccine, a positive attitude toward it, recommendations from health workers, and family support were significantly associated with vaccine uptake. Therefore, health facilities and schools should strengthen community-based health education aimed at promoting behavioral change regarding the HPV vaccine and focus on creating various training opportunities for health workers and teachers.

## Introduction

Human papillomavirus is a prevalent reproductive virus, causes various illnesses in both women and men, including precancerous lesions and cancer, and can be contracted through contact with infected skin, mucous membranes, or body fluids. Over 99% of cervical cancer cases worldwide are caused by oncogenic or high-risk STI strains, with HPV 16 and 18 being the most prevalent and dangerous [1]. Sub-Saharan Africa has the greatest regional incidence and fatality rates, with higher rates in Eastern, Southern, and Middle Africa. However, it's a preventable form of cancer through the HPV vaccine [2].

The HPV vaccine is a highly effective method for preventing HPV infections in women who have never been infected with HPV. A single dose of HPV vaccine offers similar protection against high-risk strains of HPV as two or three doses of HPV vaccination [3,4]. WHO recommends all countries introduce HPV vaccination for primary prevention of CC prioritizing the primary target group of young adolescent girls, aged 9–14 years [3]. Numerous nations have attempted to establish school-based immunization programs since their inception, and they have been astonishingly successful in raising the vaccination rates of teenage girls. However, the HPV vaccine's coverage remained low in many low- and middle-income nations, especially those whose introduction had been postponed [5,6].

With 604,000 new cases and 342,000 fatalities from cervical cancer in 2020, cervical cancer is the fourth most frequent cancer worldwide [2]. Every two minutes,

a woman loses her life to cervical cancer; most of these instances and deaths take place in low- and middle-income countries. Moreover, 90% of the cases and most of the deaths occur in developing countries where most women remain undiagnosed and have limited or no access to treatment [7].

Cervical cancer is a major reproductive health concern in Ethiopia, resulting in considerable morbidity and mortality among women between the ages of 15 and 44. The yearly incidence of the disease is 21.5 new cases and 16 deaths per 100,000 females [4,8]. Young, uneducated women in the world's poorest nations frequently seek medical attention when issues arise because they have limited access to pre-screening, treatment, advanced complications, and preventative programs and services [3].

The HPV vaccine has regrettably low global coverage; just 1.4% of eligible girls have gotten a full course of immunization, despite the fact that the vaccine has been demonstrated to cut the risk of cervical cancer [4]. Adolescent females (10–20 years old) had full course vaccination coverage in 1.2% of Africa, 1.1% of Asia, 31.1% of Europe, 19% of Latin America and the Caribbean, 35.6% of North America, and 35.9% of Oceania in 2014, according to the evidence [9]. High-income nations with successful programs, like Scotland and Taiwan, have attained >80%, according to the review study [10].

Nonetheless, a lot of sub-Saharan African nations that had delayed the introduction of the HPV vaccine still had low coverage rates; in Uganda, for example, the HPV vaccine uptake was 17.61%, but the prevalence of cervical cancer was high [11]. In a similar vein, Ethiopia's HPV vaccination coverage is low and varies, from 15% to 66.5% [12].

With assistance from GAVI, Ethiopia launched the HPV vaccine in 2018. Females aged 14 years old received the vaccine in two doses spaced six months apart. Girls who are not in school are administered the immunization through a health facility and community outreach program. When given regularly between the ages of 14 and 18, the HPV vaccine has a high rate of effectiveness in our country [13]. But for a variety of reasons, public acceptance of it has lagged [9,10,12]. Understanding the several multilayer components linked to the commencement and completion of HPV vaccination is the most important strategy to boost HPV vaccine coverage.

In order to establish a strong primary prevention program under the CC strategy and have an efficient vaccination program, it may be necessary to have a sufficient understanding of the factors that contribute to the uptake of the HPV vaccine among adolescent girls who qualify for it. The adoption of the HPV vaccine and associated factors among adolescent girls in the community, however, has not been studied in Ethiopia, particularly in the research location; the majority of prior research has been on school-going girls rather than community-dwelling girls. Furthermore, most of these previous studies on uptake of HPV vaccine were conductedonly by using a quantitative study design, and this is not enough to identify barriers and factors associated with adolescent girl's uptake of HPV vaccine. For this reason, this study used a quantitative study augmented with an explanatory sequentialmixed-method approach.

In addition, according to Ethiopia's Ministry of Education reports that 33.2% of girls in grades 6–8 and 44.6% in grades 9–12 miss school and drop out due to HPV vaccination, while 75% of secondary school girls are unrolled, and 47% of first-grade girls do not graduate fifth grade [14]. This makes it challenging to generalize the findings of earlier studies on HPV vaccination uptake and associated factors conducted in schools to the overall population. Furthermore, the prior study might not have accurately represented the level of coverage of HPV vaccination uptake. To bridge this gap, this study is therefore important to assess the uptake of HPV vaccination and associated factors among adolescent girls (14–18 years) in a general population in Merab Abaya district, Gamo Zone, southern Ethiopia.

## Methods and materials

### Study setting and period

The study was conducted from February 1 to March 30 2024 in the Merab Abaya district, located in the Gamo Zone, in the Southern part of Ethiopia. Bribir is the administrative center of Merab Abaya district, which is 50 km from Arbaminch city the capital of Gamo Zone and 554 km from Addis Ababa, the capital of Ethiopia, and 75 km from the administrative and

political center of the southern Ethiopian people region, Wolaita Sodo. The total population of Merab Abaya is 109,149. The estimated number of women in the reproductive age group (15–49) is 21,658, and the estimated number of adolescent girls (14–18) is 6,767. Merab Abaya district has 4 health centers and 27 health posts. The district also has a total of 25 kebeles, 2 urban kebeles, and 23 rural kebeles (lower-level administrations).

## Study design

A community-based cross-sectional study augmented with an explanatory sequential mixed-method approach was conducted.

## Quantitative method

A community-based cross-sectional study using structured questionnaires was employed to identify factors associated with human papilloma virus vaccination uptake among adolescent girls in Merab Abaya district Gamo Zone.

## Qualitative method

A qualitative research design with an in-depth interview and FGD was used to explore the barriers of human papilloma virus vaccination uptake among adolescent girls in Merab Abaya district Gamo zone.

## Source population

**Quantitative study.** All adolescent girls (14–18 years) in Merab Abaya district was the source population of the study.

**Qualitative method.** Adolescent girls living in Merab Abaya district, health professional, and teachers were considered as source population for qualitative data.

## Study population

**Quantitative study.** The study population was randomly selected adolescent girls aged 14–18 years at the time of data collection in selected kebeles of Merab Abaya district. The study included adolescent girls between the ages of 14 and 18 because they are the target group for HPV vaccination and the age range for the start of the HPV vaccine immunization program in the study area.

**Participant recruitment.** The study population was purposively selected adolescent girls and FGD living in the Merab Abaya district during the time of data collection. The study participants were recruited from randomly selected kebeles of study areas.

For a quantitative study, all adolescent girls (14–18 years) in Merab Abaya district was the source population of the study, and for a qualitative study, adolescent girls living in Merab Abaya district, health professional, and teachers were considered as source population.

## Eligibility criteria

**Inclusion criteria.** We have included adolescent girls aged (14–18) years from selected kebeles of the districts in a quantitative study and a purposively selected adolescent girls aged (14–18) year, health professionals, and teachers from selected kebeles of the districts in a qualitative study.

**Exclusion criteria.** Those who were critically ill, unable to respond, and not willing to participate or whose parents fail to provide assent were excluded from this study.

## Sample size determination and techniques

For the quantitative study, the required sample size was determined by using a single population proportion formula considering the following assumptions: the proportion of human papilloma virus vaccination uptake conducted in Ambo city,

central Ethiopia, which is 44% [9] with confidence level of 95%, a margin of error (d) of 5%, and a non-response rate of 10%. In addition, the design effect was calculated as 1.5 because multistage sampling technique was used as sampling procedure, and by adding 10% non-response rate, the final sample size is 626.

The sample size for 2nd objective (associated factors) was determined by using the double population proportion formula, and three key factors were taken from the previous literatures. According to the following assumptions, the sample size was computed by using Epi Info version 7.2.4.0 software (stat calc) with the following assumptions: 95% CI, level of required study power of 80%, and ratio of 1:1 (Table 1).

Based on the second objective, no sample size was greater than the sample size calculated based on single population formula. Therefore, the final sample size for this specific study was 626, taken from a single population proportion formula.

## Sampling techniques and procedures

For the quantitative study, a two-stage sampling technique was used. First, of the total of 25 kebeles/clusters present in the Merab Abaya district, 30% was selected randomly by the lottery method, which is 8 kebeles/clusters. Then, based on the estimated number of adolescent girls (14–18 years old) that the kebele/clustersholds, the total sample size was proportionally allocated to the number of selected kebeles. Finally, a simple random sampling technique using a random integer generator (RANDOM.ORG, or integer generator) was employed by using the Community Health Information System (CHIS) (health extension worker's family folder) as a sampling frame that contains the list of households that have adolescent girls in order to select study participants. If more than one eligible respondent is found in the same household, only one respondent was chosen by the lottery method (Fig 1).

The first 8 column describes the number of kebeles and adolescent girls in Merab Abaya district, which totally is 2768. The second 8 columns describe the proportional allocation of final sample to be collected from each kebeles and done by the following formula and its final sample626.

$$nj = \frac{n.Nj}{N}$$

**Where**: nj = Sample size in j kebele
n = Estimated final sample size (n = 626)
Nj = Total number of adolescent girls in j kebele
N = Total number of adolescent girls in selected kebeles of Merab Abaya District (N = 2768)

For the qualitative study, adolescent girls for in depth interview purposefully selected based on vaccination status (have not received, received one, or received two dose HPV vaccinations). The researcher interviewed a total of 16adolescent girls. Two focus group discussions (FGDs) were held with nurses, health extension workers, midwives, public health officers, and biology teachers who interacted with teenage girls and HPV vaccination campaign involvement in the Merab Abaya district. Nine people were involved in one group discussion, and seven in the other. The number of participants in interviews as well as the number of FGD ceased after information saturation (when ideas started to be repeated and no more new ideas emerged).

**Table 1. Sample size calculation using Epi info version 7.2.4 software based on associated with HPV vaccination uptake among adolescent girls (14-18 years) in Merab Abaya district, Gamo Zone, Southern Ethiopia 2024.**

| Factors | Outcomes in % | | CI | Power | AOR | S. Size | Final Sample Size(n) after NRR and design effect | Ref |
|---|---|---|---|---|---|---|---|---|
| | Exposed | Unexposed | | | | | | |
| Knowledge | 22.7 | 10.9 | 95% | 80% | 2.4 | 348 | 574 | [15] |
| Attitude | 21.9 | 8.8 | 95% | 80% | 2.9 | 268 | 442 | [15] |
| Exposure to outreach clinics | 62.8 | 13.1 | 95% | 80% | 11.2 | 36 | 59 | [16] |

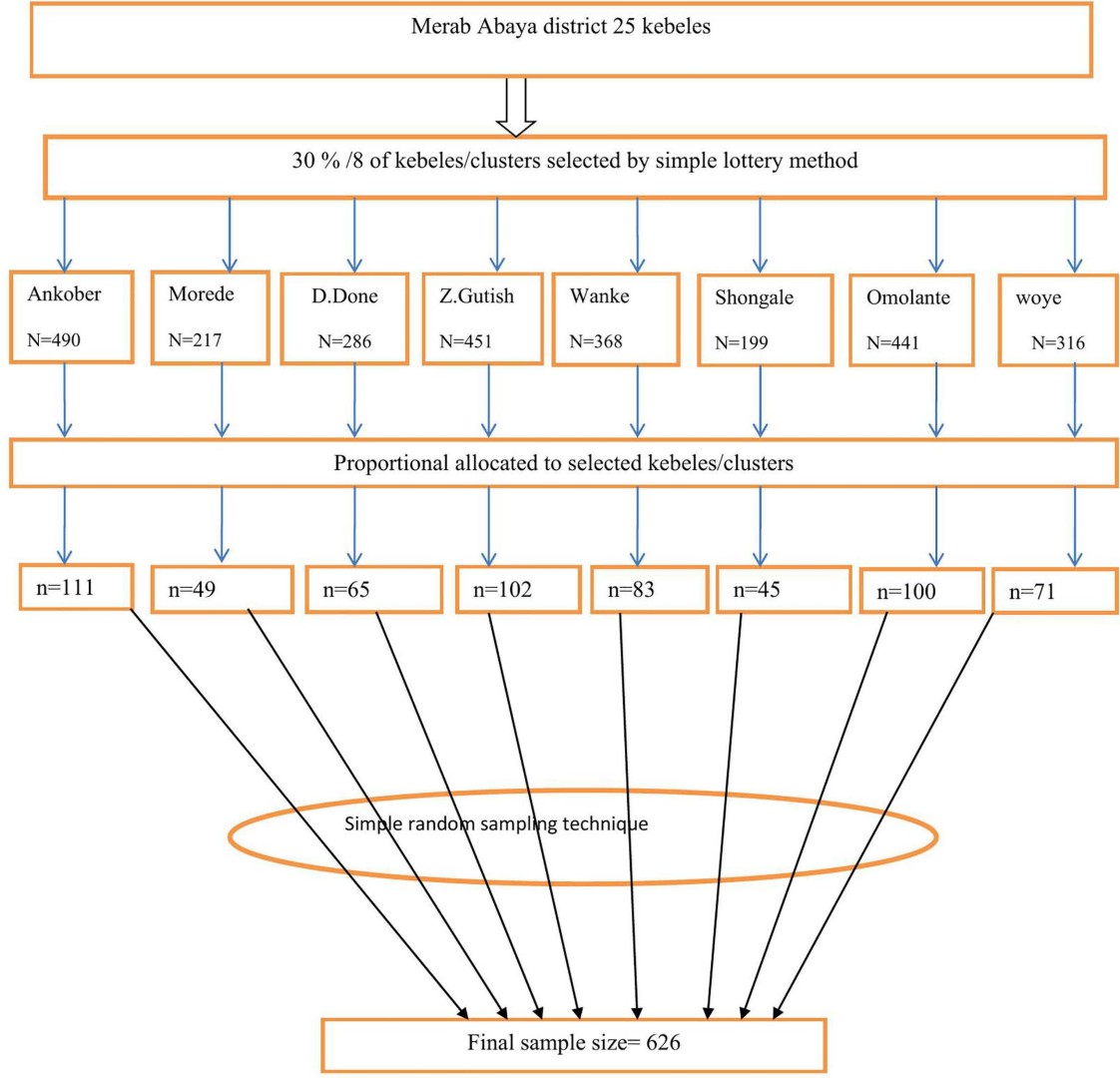

**Fig 1. Schematic presentation of the sampling procedure for human Papilloma virus vaccination uptake and associated factors among adolescent girls (14–18 years) in Merab Abaya district, Gamo zone, southern Ethiopia, 2024.**

## Study variables

**Dependent variable.** Uptake of human papilloma virus vaccination

**Independent variable.** Individual related variables: Knowledge about HPV and its vaccination, attitude about HPV and its vaccination, age, reception of childhood vaccination, schooling status, and residence.

Institutional related variables: healthcare provider's recommendation, vaccine accessibility, sensitization about the presence and availability of HPV vaccination, outreaches in the community.

Parental related variables: knowledge towards HPV and its vaccine, attitude towards HPV and its vaccine, parental support, educational level of mothers, educational level fathers, mothers occupation, family monthly income, and safety concerns of HPV.

## Operational definitions

Uptake of HPV vaccine: Refers to participants who had received at least 1 dose of HPV vaccine [17].

**Knowledge.** Knowledge was assessed by using a series of questions regarding cervical cancer, HPV infection and HPV vaccination on 10 items scale. A correct response was given a score of 1 and incorrect was scored 0.

Good knowledge about the HPV vaccine: respondents score equal to and above the 50% score was considered Good Knowledge (knowledgeable) [18].

Poor knowledge about the HPV vaccine: respondents score below 50% score was considered as Poor Knowledge (not knowledgeable) [18].

Attitude for HPV vaccine: Attitude was measured by 10 Likert scale questions in a 5- point Likert score (1strongley agree, 2 agree, 3 neutral, 4 disagree, and 5 strongly disagree), then a mean score was calculated and classified as:

Positive Attitudes toward the HPV vaccine: A participant scores equal to and above the mean score (2.45) would have a Positive Attitude [15].

Negative Attitudes toward the HPV vaccine: A participant who scores below the mean score (2.45) would have a Negative Attitude [15].

## Data collection tool and procedures

For quantitative study, the questionnaire was prepared by selecting, modifying, and adapting relevant tools regarding HPV vaccination uptake and associated factors among adolescent girls from different published research studies done on similar topics [15–18]. The questionnaire has seven sections, composed of variables related to socio-demographics, source of information, knowledge, attitude, institutional related, parental-related, and uptake of HPV vaccination. The questionnaire was prepared in English, translated to the Amharic language for data collection, and finally translated back to English to check its consistency. Data was collected using a pretested, structured questionnaire through a face-to-face interview. Eight trained diploma nurses were recruited for data collection. The data collectors were trained for two days on how to collect, fill, and handle the data according to the objective of the study, the contents of the questionnaire, issues related to the confidentiality of the responses, and the rights of the respondents.

The data for the qualitative study, a semi-structured interview guide was prepared and used in both the in-depth interview and focus group discussion. The principal investigator was taken video-based lectures to upgrade his capacity on how to approach participants, how to handle them during interviews, how to probe, and how to ask sensitive questions. The topic guide addressed all issues related to barriers with human Papilloma virus uptake and other issues relevant to the objectives of the study. The principal investigator had conducted the discussions, while one assistance co-facilitator had assigned during the FGDsto manage discussion time and facilitate recording. IDI and FGD were guided by an experienced person fluent in the local language (Gammotho) and English with the researcher. The interview had been recorded on tapes, which were later transcribed. Detailed field notes had been taken.

## Data processing and analysis

**For quantitative study**, data was coded, entered into Epi Data version 4.6, and exported to SPSS version 26 for analysis, with descriptive statistics reported as frequencies and percentages, and presented in tables and graphs. Both bivariate and multivariate logistic regression analyses were used to determine the association of each independent variable with the dependent variable. A multi co linearity test between independent variables was checked using the variance inflation factor and all variables was less than 2.2, and all variable has tolerance greater than 0.45.

All independent variables with a p-value less than 0.25 from the bivariable logistic regression analysis were entered into the multivariable logistic regression analysis to control the possible effect of confounders. All the assumptions for binary logistic regression, i.e., Model Goodness-of-Fit, were checked by Hosmer and Lemeshow test at p-value of0.69.

A significant association was obtained at an adjusted odds ratio (AOR) with a 95% confidence interval (CI) and p-value less than 0.05 for interpretation. Different frequency tables with the AOR, graphs, charts, and descriptive summaries were used to describe the study variables.

**For qualitative study**, prior to analysis, the voice records were transcribed word by word in Amharic language as soon as possible after the FGDs as well as IDI and notes were also organized. Then, the data were translated into English language. After that, the translated data were exported into Open Code4.02 software for analysis. Then, the data were broken into many sentences and codes were formed, and then thematic content analysis was conducted for this study. Finally, an original verbatim quotation from participants was used to support or augment the data from quantitative analysis.

## Data quality assurance

The data quality for the quantitative study was assured by different methodologies. Adequate training was given to data collectors and supervisors on the data collection tool and data collection procedure. The questionnaire was pre-tested a week before the actual data collection days on 5% (31) of the sample size, and any necessary modifications was made accordingly. The pre-test was done in Boreda district, Kebele, which has a similar socio-demographic background but is not part of the study area, to ensure its validity, and necessary corrections was made based on the results of the pre-test before the actual data collection (i.e., 31 adolescent girls).

Data collectors was informed and encouraged to conduct interviews at attractive and convenient times. Regular and continuous follow-up was made by the principal investigator to monitor the quality of the data collection process, and every completed questionnaire was checked daily for completeness and consistency, and feedback was given to the data collectors.

To assure data quality for a qualitative study, two days of intensive reading and understanding of the study objectives, informed consent, confidentiality of information, and interview techniques were done by researchers. Trustworthiness was ensured through prolonged engagement by establishing enough contact with study participants to get an adequate understanding of the concept or to ensure the rigor of the study's credibility, transferability, dependability, and confirmability were maintained.

To ensure credibility, the researcher clarified the method of data analysis, contents of the checklist, and any other issues at the time of IDI and FGD for participants in order to validate the results. To certify transferability, sixteen adolescent girls recruited from study area participated in IDI and two sessions' of FGDs. Dependability was strengthened by presenting an in-depth description of the processes within the study, other issues that occur during the interview and over the course of the study. To warrant confirmability, an investigator had described the purpose of the study, an audio record, the objective of the study, norms for discussion, time allotment, every step of data analysis, and ethical related issues for IDI and FGD participants, and the data were preserved.

## Declaration

### Ethics approval and consent to participate

The ethical clearance was obtained from the Ethical Review Board Committee of Wolaita Sodo University, School of Public Health through an ethical letter with protocol number CHSM/ERC/01/16. After that, informed written consent was obtained from every study participant prior to data collection, and a letter of cooperation was submitted to the Merab Abaya district chairman of each selected kebele. Additionally, respondents were told that they might decline the questionnaire and that any information given would be treated with confidentiality to preserve their privacy. Above all, each step of this study was carried out in accordance with the Declaration of Helsinki's ethical guidelines for medical research on human subjects.

**Quantitative data.** The advantages and purposes of the study were explained to study participants. The data were collected after providing written consent to each study participant to participate in the study. Respondents were also informed that they had the right to decline the questionnaire and all the information provided was handled in a confidential manner to keep the privacy of the respondents. Above all, this study was entirely conducted as per the Declaration of Helsinki ethical principles for medical research on human subjects.

**Qualitative data.** Major ethical principles such as beneficence, non-maleficence, autonomy, and justice were followed to ensure no harm would come to the participants throughout this study.

**Beneficence and non-maleficence.** The proposed study findings should benefit and cause no harm to the participants and society at large. The researcher had aimed to raise HPV vaccination rates and enhance teenage girls' overall health outcomes. Privacy and confidentiality were maintained at all times, all findings were portrayed in a confidential manner no personal or identifiable information was recorded or printed in the study. Electronic voice recorder was used during IDIs and focused group discussion and the voice was transcribed word by word into original language which was Amharic language and translated into English language, but no names were recorded during the discussion process. Once transcribed and translated, the data were stored in password protected folders with restricted access and stored on an external hard drive which only the researcher had an access to. The data held on a computer were accurate, and it also allows the participants to view the information regarding themselves and to correct any errors if they so wish.

**Autonomy.** The investigator had respected the human right of free choice and ensured informed consent is completed before carrying out any discussion with participants. The researcher had ensured a regular review of what the participants have given consent to is carried out. All participants were reassured that the option to refuse to participate and withdraw from the research at any time without penalty or effects was maintained.

**Justice.** All participants' experiences, opinions and perceptions were depicted as they have done so in the discussion, no false information or accusations were included in the final research report.

**Anonymity and confidentiality.** The anonymity and confidentiality of the participants was preserved by not revealing their names and identity in the data collection, analysis and reporting of the study findings. Privacy and confidentiality of the discussion environment was managed carefully during discussion, data analysis and dissemination of the findings.

**Information sheet and consent form.** Information about the entire study and written consent was obtained from participants prior to IDIs and focus group discussions (FGDs).

## Results

### Socio demographic characteristics of respondents

A total of 626 adolescents were included in this study, and approximately 601 respondents participated, yielding a response rate of 96%. The mean age of the study participants was 15.76 years (SD ± 1.35), and the highest proportions of the age were found within the age range of 16–18 years (60.7%). Of the study participants, 236 (39.3%) were followers of the Protestant religion. About 473 (78.7%) of the respondents were in school, whereas 286 (47.6%) of them were in grades 7–9. Regarding parental educational status, more than a third, 207 (37.4%) and 213 (35.4%) of the participant's fathers and mothers had no formal education, respectively. The majority, 231 (38.4%) of the participants, stated that their family's income was 4000–7000 ETB per month (Table 2).

### Source of information about the HPV vaccine

Out of a total of 601 participants, 422 (70.2%) heard of the HPV vaccine. From those, 164 (38.9%) heard the HPV vaccine in schools or from teachers, 98 (23.2%) heard it from health professionals, 67 (15%) heard it on TV or radio, 45 (10.7%) heard it from peers, 34 (8%) heard it on the internet or social media, and 14 (3.3%) heard it from family (Fig 2).

**Table 2. Socio-demographic characteristics of the adolescent girls (14–18 years) in Merab Abaya district, Gamo Zone, Southern Ethiopia, 2024.**

| Variable | Category | Frequency | Percentage |
|---|---|---|---|
| Age | 14 −15 | 236 | 39.3 |
| | 16–18 | 365 | 60.7 |
| Religion | Orthodox | 232 | 38.6 |
| | Protestant | 236 | 39.3 |
| | Muslim | 110 | 18.3 |
| | Other (specify)* | 23 | 3.8 |
| Ethnicity | Gamo | 439 | 73.0 |
| | Wolaita | 145 | 24.1 |
| | Gofa | 10 | 1.7 |
| | Other specify** | 7 | 1.2 |
| Schooling status | In school | 473 | 78.7 |
| | Out of school | 128 | 21.3 |
| Level of education | 4–6 | 27 | 4.5 |
| | 7–9 | 286 | 47.6 |
| | 10–12 | 160 | 26.6 |
| School mini-media | Yes | 383 | 63.7 |
| | No | 90 | 15.0 |
| Residence | Urban | 107 | 17.8 |
| | Rural | 494 | 82.2 |
| Fathers education | Unable to read and write | 207 | 34.4 |
| | Able to read and write | 146 | 24.3 |
| | Primary education | 77 | 12.8 |
| | Secondary education | 78 | 13.0 |
| | College diploma and above | 93 | 15.5 |
| Mothers education | Unable to read and write | 213 | 35.4 |
| | Able to read and write | 120 | 20.0 |
| | Primary education | 90 | 15.0 |
| | Secondary education | 93 | 15.5 |
| | College diploma and above | 85 | 14.1 |
| Mothers occupation | Government employed | 80 | 13.3 |
| | Private employed | 24 | 4.0 |
| | Merchant | 68 | 11.3 |
| | Farmer | 15 | 2.5 |
| | House wives | 414 | 68.9 |
| Father occupation | Government employed | 95 | 15.8 |
| | Private employed | 82 | 13.6 |
| | Merchant | 112 | 18.6 |
| | Farmer | 312 | 51.9 |
| Monthly income | < 4000ETB | 212 | 35.1 |
| | 4000–7000ETB | 231 | 38.9 |
| | >7000ETB | 159 | 26.5 |

Others* = Apostolic, Others** = Amhara, Oromo, ETB: Ethiopian Birr.

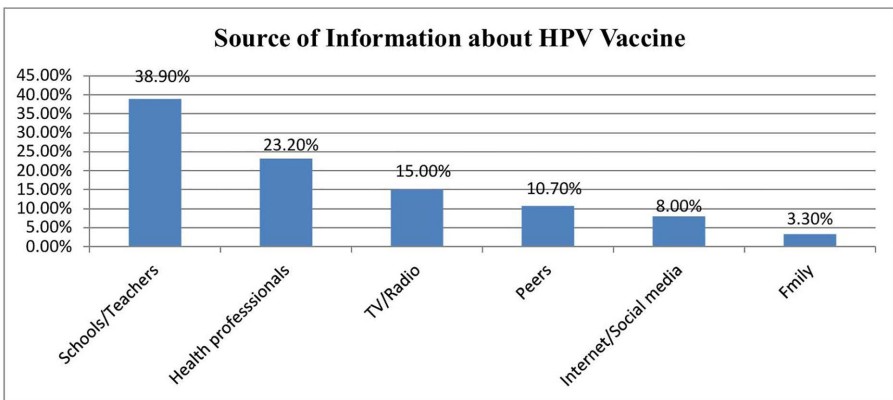

**Fig 2. Source of information on HPV vaccination adolescent girls in Merab Abaya district, Gamo Zone, Southern Ethiopia, 2024.**

## Knowledge about HPV and vaccination

In this study, ten items were used to assess knowledge of HPV and vaccination. The lowest score was 0 and the highest was 10, and an adolescent girl's equal to or above 50% score was considered good knowledge, whereas an adolescent girl's knowledge score below 50% was considered poor knowledge.

Overall, 303 (50.4%) adolescent girls have good knowledge about the HPV vaccine. More than two-thirds of participants 493(82%) knew the HPV vaccination given for the prevention of cervical cancer. More than half of the participants, 342 (56.9%), knew the HPV vaccine should be given before the first sexual intercourse, and 382 (63.6%) knew the HPV infection could be transmitted through sexual contact. Also, more than half of respondents 354(58.9%) knew the recommended age range for HPV vaccine vaccination, which is between the ages of 9 and 14 years(Table 3).

Corresponding to survey findings, IDI participants most frequently mentioned adolescent girls have poor knowledge about uptake of HPV vaccine, as illustrated below:

*"I have never been vaccinated. My reason for not getting vaccinated is because I don't know much about vaccination."* **(IDIP-11, 18 years old, unmarried, not educating girl)**

## Attitude towards HPV vaccine

The attitude of the participants was measured using 10 items of Likert scale data, then a mean score was calculated, and 10 questions were recorded as follows: positive attitude and negative attitude. The overall positive attitude towards the HPV vaccine among participants was 204 (33.9%).

According to this study, 45.1% of respondents strongly agreed that the HPV vaccine can prevent cervical cancer, and about 226 (37.6%) respondents agreed that they are at risk for HPV infection in the future. On the other hand, about 223 (37.1%) of respondents strongly agreed that screening for HPV infection helps in the early detection of cervical cancer.

Nearly half of the respondents (41.6%) strongly agreed that cervical cancer is a deadly disease, whereas about 243 (40.3%) disagreed that the HPV vaccine is safe and effective at preventing cervical cancer. Also, nearly half of the respondents (44.9%) strongly agreed that health professional counseling affects decisions as to whether or not to receive vaccinations. About 221 (36.8%) strongly agreed that it was better to be vaccinated before becoming sexually active (Table 4).

**Table 3. Adolescent girls (14-18 years) level of knowledge about HPV vaccination in Merab Abaya district, Gamo Zone, Southern Ethiopia, 2024.**

| Knowledge items | Category | Frequency | Percentage |
|---|---|---|---|
| Does HPV vaccine prevent cervical cancer? | Given for prevention of cervical cancer | 493 | 82 |
| | Given for prevention of malaria | 7 | 1.2 |
| | Given for prevention of HIV | 9 | 1.5 |
| | I do not know | 92 | 15.3 |
| Did you know the schedule for a human papilloma virus vaccination? | 6 months | 388 | 64.6 |
| | 6 years | 18 | 3.0 |
| | 1 month | 33 | 5.5 |
| | I do not know | 162 | 27.0 |
| Should a human papilloma virus vaccine be given prior to the first sexual intercourse? | Yes | 342 | 56.9 |
| | No | 33 | 5.5 |
| | I do not know | 226 | 37.6 |
| Who should get the HPV vaccination? | Women | 297 | 49.4 |
| | Men | 16 | 2.7 |
| | Both men and women | 207 | 34.4 |
| | I do not know | 81 | 13.5 |
| How many doses are recommended for a human papilloma virus vaccine? | Once | 135 | 22.5 |
| | Twice | 179 | 29.8 |
| | More than two | 151 | 25.1 |
| | I do not know | 136 | 22.6 |
| Who are infected with human papilloma virus? | Women | 218 | 36.3 |
| | Men | 23 | 3.8 |
| | Both men and women | 235 | 39.1 |
| | I do not know | 125 | 20.8 |
| What age range is recommended for vaccination against human papilloma virus infection? | 9–14 years | 354 | 58.9 |
| | 18–21 years | 45 | 7.5 |
| | All ages | 41 | 6.8 |
| | I do not know | 161 | 26.8 |
| Possible to have cervical cancer screening after HPV vaccination? | Yes | 358 | 59.6 |
| | No | 44 | 7.3 |
| | I do not know | 199 | 33.1 |
| HPV infection can be transmitted through? | Sexual Contact | 382 | 63.6 |
| | Aerosol | 87 | 14.5 |
| | I don't know | 132 | 22.0 |
| All HPV infection is healed by itself without treatment? | Yes | 247 | 41.1 |
| | No | 42 | 7.0 |
| | I do not know | 312 | 51.9 |

Corresponding to survey findings, IDI participants articulated those adolescent girls have mistrust on vaccine because acknowledged a widespread fear that they could render girls infertile, cause them to become members of the 666, as illustrated below:

*"… people think, "It's bad for girls in the future, it can prevent them from having children, it's related with 666, it can cause excessive sex drive and that can prevent them from learning properly." Even if we are told that we are not afraid*

**Table 4. Adolescent girls (14-18 years) Attitude level about HPV vaccination in Merab Abaya district, Gamo Zone, Southern Ethiopia, 2024.**

| Attitude status | Strongly disagree | Disagree | Neutral | Agree | Strongly agree |
|---|---|---|---|---|---|
| HPV vaccine can prevent cervical cancer. | 16 (3.2%) | 66 (11.0%) | 96 (16.0%) | 206 (34.3%) | 214 (35.6%) |
| Risk of getting an HPV infection in the future. | 13 (2.2%) | 113 (18.8%) | 124 (20.6%) | 226 (37.6%) | 125 (20.8%) |
| Cervical cancer is a deadly disease | 20 (3.3.3%) | 56 (9.3%) | 90 (15.0%) | 185 (30.8%) | 250 (41.6%) |
| Regular screening for HPV infection helps early detection of cervical cancer. | 12 (2%) | 76 (12.6%) | 104 (17.3%) | 186 (30.9%) | 223 (37.1%) |
| The vaccine's side effects are reasonable and will not prevent me from receiving the vaccine | 46 (7.7%) | 87 (14.5%) | 84 (14.0%) | 191 (31.8%) | 193 (32.1%) |
| Better to be vaccinated before becoming sexually active | 21 (3.5%) | 66 (11.0%) | 96 (16.0% | 197 (32.8%) | 221 (36.8%) |
| Health professional counseling affects your decision as to whether or not to receive vaccinations. | 8 (1.3%) | 58 (9.7%) | 86 (14.3%) | 179 (29.8%) | 270 (44.9%) |
| More information on HPV and its vaccine will be needed before I take the vaccine. | 11 (1.8%) | 48 (8.0%) | 87 (14.5%) | 184 (30.6%) | 271 (45.1%) |
| The HPV vaccine is safe and effective | 11 (1.3%) | 242 (40.3%) | 138 (23.0) | 160 (26.6%) | 50 (8.3%) |
| Not easy to find the HPV vaccine | 55 (9.2%) | 35 (5.8%) | 215 (35.8%) | 104 (17.3%) | 192 (31.9%) |

*of that gossip, but sometimes that gossip scares us. They say that it is 666. This is meant to destroy people. It is meant to harm future people." (IDIP-4, 17 years old unmarried, Grade 9th girl)*

### Institutional related factors of respondents

The finding showed that more than three-fourths of respondents (75.4%) do not have HPV vaccination services available in the nearby government health facility or school of their choice. Almost two-thirds of the respondents (74.5%) have not obtained an HPV vaccine from an outreach clinic or health post, but 587 (97.7%) of the respondents have not found a government health facility, school, or outreach/health post offering the HPV vaccination service on a regular or consistent schedule.

About two-thirds of respondents (68.7%) have received advice from any health workers about getting vaccinated for HPV, whereas 527 (87.7%) of respondents have not had healthcare workers in their community conduct health education campaigns regarding the HPV vaccine or cervical cancer (Table 5).

Corresponding to survey findings, IDI participants mentioned that adolescent girls have lack of information provision from health professionals and non-regular provision of vaccine, as illustrated below:

*"Earlier it was given attention, but now it is less. It has reduced a lot. In the past, health extension and health professionals would come and teach while students were lining up. Now, it has reduced."(IDIP-1, 14 years old unmarried, Grade 8th girl)*

*"Another is that the vaccine comes and is given and then there is no continuity. Its fine if that's not the case at all. Such an interruption would appear to be another problem in itself." (P-4, 17 years old unmarried, Grade 9th girl)*

### Parental related factors of HPV and vaccination

The majority of respondents (62.1%) have not received HPV vaccinations during childhood, but more than half of the respondents (57.7%) have parents aware of cervical cancer, and cervical cancer vaccines prevent HPV. Also, more than two-thirds of respondents (71.0%) do not believe that the HPV vaccine is safe and effective; on the other hand, 392 (65.2%) reported that their family or guardians supported them in taking the HPV vaccine (Table 6).

**Table 5. Health system related factors about HPV vaccination among Adolescent girls (14-18 years) in Merab Abaya district, Gamo Zone, Southern Ethiopia, 2024.**

| Variables | Category | Frequency | Percent |
|---|---|---|---|
| HPV Vaccination services available in the nearby government health facility/school of your choice | Yes | 148 | 24.6 |
| | No | 453 | 75.4 |
| Obtain an HPV vaccine from an outreach clinic/health post | Yes | 153 | 25.5 |
| | No | 448 | 74.5 |
| The government health facility, school and outreach/health post offering the HPV vaccination service on a regular or consistent schedule? | Yes | 14 | 2.3 |
| | No | 587 | 97.7 |
| Estimated distance from health facility or school (service site): less than 2 km (walk able) | Yes | 226 | 37.6 |
| | No | 375 | 62.4 |
| Received advice from any health workers about getting vaccinated for HPV? | Yes | 413 | 68.7 |
| | No | 188 | 31.3 |
| Health facilities and schools providing adequate information about the HPV vaccine for you and your parents in order to make an informed decision | Yes | 128 | 21.3 |
| | No | 473 | 78.7 |
| Healthcare workers in your community conduct health education campaigns regarding the HPV vaccine or cervical cancer | Yes | 74 | 12.3 |
| | No | 527 | 87.7 |

**Table 6. Parental-related factors of HPV vaccine among adolescent girls (14 - 18 years) in Merab Abaya district, Gamo Zone, Southern Ethiopia, 2024.**

| Variables | Category | Frequency | Percent |
|---|---|---|---|
| Receive vaccinations during childhood | Yes | 228 | 37.9 |
| | No | 373 | 62.1 |
| Parents are aware of cervical cancer and cervical cancer vaccines available to prevent it | Yes | 347 | 57.7 |
| | No | 254 | 42.3 |
| Parents believe that the HPV vaccine is safe and effective | Yes | 174 | 29.0 |
| | No | 427 | 71.0 |
| Parents ever discussed or talked to you about HPV vaccination | Yes | 231 | 38.4 |
| | No | 370 | 61.6 |
| Family or guardians support you taking the HPV vaccine? | Yes | 392 | 65.2 |
| | No | 209 | 34.8 |

## Uptake of HPV vaccine

More than half of the respondents (53.9%) (95% CI: 49.9%–57.9%) had ever received any dose of HPV vaccine; of them, 164 (27.3%) had received one dose and 160 (26.6%) had received two doses. Out of the total participants, 277 (46.1%) were not vaccinated against HPV. The main reasons for not receiving the HPV vaccine were lack of parental support 60 (10%), fear of needle injection of the HPV vaccine 58(9.5%), not being informed about the vaccine 47(7.8%), lack of access to the HPV vaccine 35(5.8%), and lack of social support or rumors about the vaccine 27(4.3%) (Fig 3).

## Factors associated with uptake of human papilloma virus vaccination

For describing factors associated with the uptake of HPV vaccine among adolescent girls in Merab Abaya district, Gamo Zone, the first variables were checked for multicollinearity, and the variables that passed the test were entered into a bivariable logistic regression analysis. Variables with a p-value of less than 0.25 were entered into a multivariable logistic regression analysis.

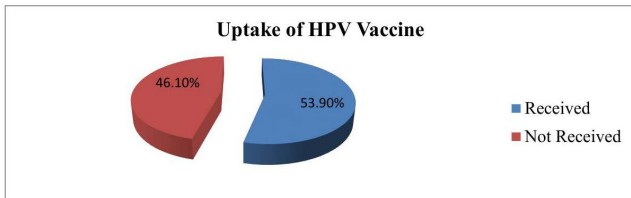

**Fig 3.** Uptake of the HPV vaccination among adolescent girls in Merab Abaya district, Gamo Zone, Southern Ethiopia, 2024.

A total of fifteen variables were used for the bivariable logistic regression analysis. Among these, all of the variables (age, schooling status, residence, fathers educational level, mothers educational level, monthly income, knowledge, attitude, vaccine availability in a near government health facility, vaccine obtained from an outreach clinic, received advice from health workers, received vaccination during childhood, aware of the HPV vaccine, parents ever discussed it, and family or guardians support for receiving the HPV vaccine) had an association with HPV vaccination uptake in bivariable logistic regression analysis at a p-value of < 0.25.

In multivariate logistic regression, some factors were depicted as significant (P-value less than 0.05) factors that affect the uptake of HPV vaccine among adolescent girls in Merab Abaya district, Gamo Zone. Those factors are Age of participants, the current schooling status of the participants; knowledge towards HPV vaccine; attitude towards HPV vaccine; receiving advice from health workers; and family or guardian's support to receive the HPV vaccine.

The finding shows that Adolescent girls who were at the age of 16 up to 18 years old were 1.8 times [AOR (95% CI) 1.8 (1.14–2.9)] more likely to uptake the HPV vaccination than those who were at the age of 14 up to 15 years old.

This study found that adolescent girls who were currently in school were 2.3 times [AOR (95% CI) 2.3 (1.22–4.4)] more likely to uptake the HPV vaccination than those who were current out of school.

IDI participants explained that adolescent girls with out of school had low level of understanding about the uptake of PV vaccine, saying

*"I have never been vaccinated. My reason for not getting vaccinated is because I don't understand much about vaccination."*

Participants who were well knowledgeable about HPV vaccination were 3.4 times [AOR (95% CI) 3.4 (2.14–5.38)] more likely to uptake the HPV vaccination than less knowledgeable participants. The participants said poor knowledges are the reason for not vaccinating HPV, as illustrated in (Table 7):

*"Now, most of our students are afraid, they are very afraid to vaccinate, but they are very afraid because they do not know the advantages and disadvantages"* **(A 24 Years old, FGD1, 4 year experienced public health officer)**

Participants who had a positive attitude towards the HPV vaccine were 1.7 times [AOR (95% CI) 1.7 (1.024–2.78)] more likely to uptake the HPV vaccine than those who had a negative attitude toward it. Furthermore, respondents who received advice from health workers were 3.8 times [AOR (95% CI)3.8 (2.25–6.5)] more likely to uptake the HPV vaccine than those who had not received advice from health workers.

Also, respondents who got support from their family or guardians to get the HPV vaccine were 7.1 times [AOR (95% CI) 7.1 (3.966–12.6)] more likely to uptake the HPV vaccine than those who did not receive support (Table 7).

IDI participants reported that the absence of parental support challenges of the uptake of HPV vaccine, as illustrated below:

**Table 7. Factors associated with uptake of human papilloma virus vaccination among Adolescent girls (14-18 years) in Merab Abaya district, Gamo Zone, Southern Ethiopia, 2024.**

| Variables with category | | HPV uptake | | COR with 95% CI | AOR with 95% CI | P-Value |
|---|---|---|---|---|---|---|
| | | Received n (%) | Not Received n (%) | | | |
| Age | 14–15 | 120 (20) | 116 (19.2) | 1 | 1 | |
| | 16–18 | 204 (34) | 161(26.8) | 1.2(0.9-1.7) | 1.8(1.1-2.9) | 0.013* |
| Current schooling status | In school | 298 (49.6) | 175 (29) | 6.7 (4.2-10.7) | 2.3(1.2-4.4 | 0.01* |
| | Out school | 26 (4.4) | 102 (17) | 1 | 1 | |
| Residence | Urban | 74 (12.3) | 33 (5.5) | 2.2 (1.4-3.4) | 0.9(0.5-1.6) | 0.64 |
| | Rural | 250 (41.6) | 244 (40.6) | 1 | 1 | |
| Fathers educational level | Unable to read and write | 76 (12.6) | 131(21.8) | 1 | 1 | |
| | Able to read and write | 67 (11.2) | 79 (13.1) | 1.5 (1.0-2.3) | 1.1(0.6-1.7) | 0.92 |
| | Primary education | 45(7.5) | 32 (5.3) | 2.4 (1.4-4.1) | 1.6(0.7-3.4) | 0.25 |
| | Secondary education | 57 (9.5) | 21(3.5) | 4.7 (2.6-8.3) | 1.5(0.7-3.4) | 0.3 |
| | College diploma and above | 79 (13.2) | 14 (2.3) | 9.7 (5.2-18.3) | 2(0.8-5.2) | 0.17 |
| Mothers educational level | Unable to read and write | 76 (12.6) | 137 (22.8) | 1 | 1 | |
| | Able to read and write | 66 (11) | 54 (9) | 2.2 (1.4-3.5) | 1.3(0.7-2.3) | 0.48 |
| | Primary education | 51(8.5) | 39 (6.5) | 2.4 (1.4-3.9) | 1.5(0.7-3.1) | 0.27 |
| | Secondary education | 63(10.5) | 30 (5) | 3.8 (2.3-6.4) | 1.7(0.8-3.5) | 0.17 |
| | College diploma and above | 68 (11.3) | 17 (3) | 7.2 (3.9-13.2) | 1.8 (0.7-4.4) | 0.19 |
| Monthly income | < 4000 | 86 (14.3) | 125 (20.8) | 1 | 1 | |
| | 4000–7000 | 123 (20.5) | 108 (18) | 1.7(1.1-2.4) | 1.1 (0.6-1.8) | 0.87 |
| | >7000 | 115(19.1) | 44 (7.3) | 3.8 (2.4-5.9) | 0.9(0.4-1.9) | 0.75 |
| Knowledge | Poor knowledge | 97 (16.2) | 201 (33.4) | 1 | 1 | |
| | Good knowledge | 227 (37.8) | 76 (12.6) | 6.2 (4.3-8.8) | 3.4 (2.1-5.4) | 0.000** |
| Attitude | Negative attitude | 224 (37.3) | 179 (29.8) | 1 | 1 | |
| | Positive attitude | 100 (16.6) | 98 (16.3) | 0.8 (0.6-1.1) | 1.7 (1.0-2.8) | 0.04* |
| Vaccine availability in near government health facility | Yes | 93 (15.4) | 55 (9.2) | 1.6 (1.1-2.4) | 1.4 (0.8-2.4) | 0.25 |
| | No | 231(38.4) | 222 (37) | 1 | 1 | |
| Vaccine obtain from outreach clinic | Yes | 96 (16) | 57 (9.5) | 0.8 (0.9-1.7) | 1.4 (0.9-2.2) | 0.17 |
| | No | 295 (49) | 153 (25.5) | 1 | 1 | |
| Received advice from health workers | Yes | 266 (44.3) | 147 (24.5) | 4 (2.8-5.9) | 3.8 (2.3-6.5) | 0.000** |
| | No | 58 (9.7) | 130 (21.6) | 1 | 1 | |
| Receive vaccination during child hood | Yes | 144 (24) | 84 (14) | 1.8 (1.3-2.6) | 1.3 (0.8-2.1) | 0.26 |
| | No | 180 (30) | 193 (32) | 1 | 1 | |
| Aware of HPV vaccine | Yes | 255 (42.4) | 92 (15.3) | 7.4 (5.2-10.7) | 1.4 (0.8-2.4) | 0.2 |
| | No | 69 (11.5) | 185 (30.8) | 1 | 1 | |
| Parents ever discussed | Yes | 162 (27) | 69 (11.4) | 3 (2.1-4.3) | 1.3 (0.8-2.2) | 0.27 |
| | No | 162 (27) | 208 (34.6) | 1 | 1 | |
| Family/ guardians support to receive HPV vaccine | Yes | 289 (48) | 103 (17) | 14 (9.1-21.4) | 7.1(4.0-12.6) | 0.000** |
| | No | 35 (6) | 174 (29) | 1 | 1 | |

Key: 1 = References, *Significant at p-value < 0.05, **p-value <0.001, COR = Crude Odd Ratio, AOR = Adjusted Odd Ratio, CI = Confidence Interval.

*"The family says, "Don't just go and take it, hear what it's good for and get vaccinated. Don't just take it without understanding what it is." My family says that this thing is bad, they warn us not to take it, and they don't let us. They tell us to "understand, know why it is useful, and take it"."***(P-10, 15 years old not married, grade 8th girl)**

### In-depth interview and focused group discussion (FGD) results

In-depth interviews (IDIs) were conducted on 16 adolescent girls participants, with an ages ranged from 14 to 18 years old. The data were organized into three themes based on the data obtained from the IDIs.

Two focus group discussions (FGDs) were held with nurses, health extension workers, midwives, public health officers, and biology teachers who interacted with teenage girls in the Merab Abaya district. Nine people were involved in one group discussion, and seven in the other. Participants in the FGD included one biology teacher, one pharmacy technician, one midwife, four nurses, four public health officers, and four HEW. Professionals' experience spans from four to sixteen years.

### Barriers to uptake of human papilloma virus vaccine

In the study, 30 codes, 9 subthemes, and 3 themes were identified relating to the perception and barriers of Human papilloma virus vaccine among adolescent girls. The themes that emerged were: Awareness and experience of FP (use of modern contraception), Perception of MCs use, barriers to human papilloma virus vaccine uptake and Recommendation for better uptake. The themes and subthemes are shown as follows (Fig 4).

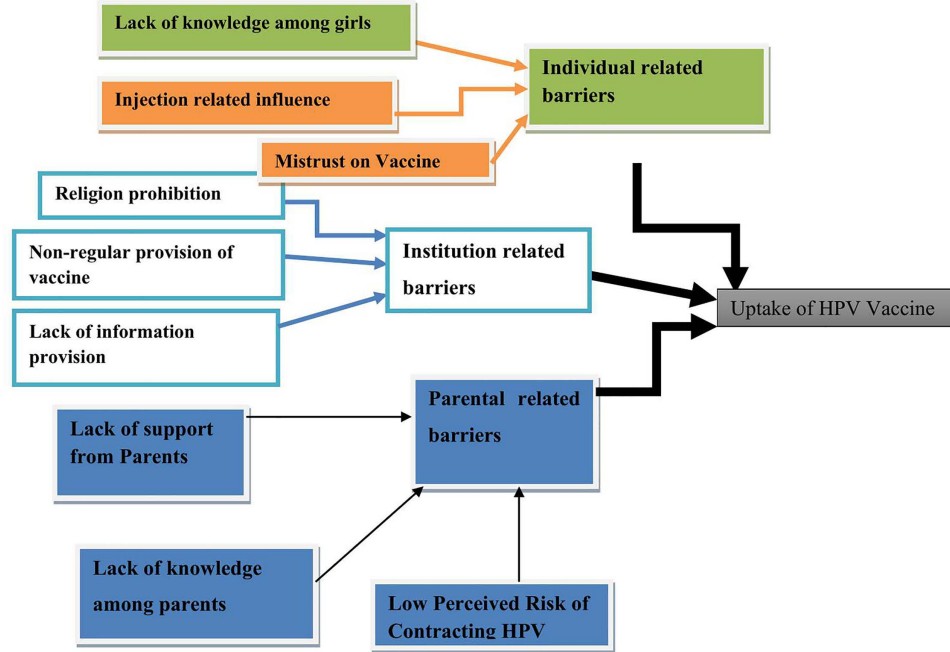

**Fig 4. Presentation of the themes and sub-themes of Adolescent girls' perception and barriers to Human Papilloma Virus Vaccine in Merab Abaya district, Gamo Zone, Southern Ethiopia, 2024.**

**Theme 1 Individual related barriers**

**1.1.    Lack of knowledge among girls**

While the majority of the girls interviewed in the qualitative part knew the HPV vaccine, some revealed that they did not know the vaccine. The others complained about not receiving enough information from their healthcare provider or school, which resulted in their lack of knowledge and non-uptake of vaccine.

A 30-year-old public health officer who had 4 years experience participated in FGD 1 expressed this notion as follows:

*"Now, most of our students are afraid, they are very afraid to vaccinate, but they are very afraid because they do not know the advantages and disadvantages."*

IDIP-16, 16 years old unmarried grade 11th girl participated in IDI expressed this notion as follows:

*"Half of the girls are vaccinated, and half of the girls are not. They (professionals) teach health education but they don't teach it well, nobody teaches us enough."*

**1.2.    Mistrust on Vaccine**

Adolescent girls who voiced doubts about vaccinations typically talked about how terrible time is and how they can't trust anyone or anything during this scary period. Adolescent girls expressed mistrust towards vaccinations and acknowledged a widespread fear that they could render girls infertile, cause them to become members of the 666, and ultimately be used to lower the population.

IDIP-3, 18 years old unmarried Grade 11thgirl participated in IDI expressed this notion as follows:

*"Students talk saying that this vaccine is not good, it may harm girls in the future, and the government has thought of something else and brought it, so we will not take the injection. They say that if it was something useful, they would not give it away for free."*

**1.3.    Injection related influence**

One of the barriers to non-uptake of vaccination for the majority of participants was their fear of needles.
IDIP-12, 18 years old unmarried Grade 12th girl participated in IDI expressed this notion as follows:

*"I have never been vaccinated. I am very afraid of needles. When I see others getting injections, I run away thinking that they are going to inject me too. When we girls live together, they(boys) talk a lot that it (vaccine) will make us infertile, so I was scared and never went near the vaccine. In addition to being afraid of needles, I don't think it will catch me because I don't have close contact, so I convinced myself that I wouldn't get injected.*

The concern over the potential short-term negative effects of vaccination uptake was the other barrier mentioned here. After receiving the injection, some girls experience dizziness, and others decide not to receive the vaccine after learning what it is meant to do.

A 32-year-old BSc Nurse professional who had 5years experience participated in FGD 2 expressed this notion as follows:

*"When adolescent girls experience an adverse effect following immunization (AEFI) such as pain, swelling, and dizziness, there is a situation where the girls are worried about taking it."*

**Theme 2 Institution related barriers**

**Subtheme 2.1 Lack of information provision.** Some of the adolescent girls reported that not getting specific vaccine advice from health professionals as one of the institutional barriers to receiving the HPV vaccine among adolescent girls.

IDIP-1, 14 years old unmarried Grade 8th girl participated in IDI expressed this notion as follows:

*"Earlier it was given attention, but now it is less. It has reduced a lot. In the past, health extension and health professionals would come and teach while students were lining up. Now, it has reduced."*

IDIP-16, 16 years old unmarried Grade 11th girl participated in IDI expressed this notion as follows:

*"Half of the girls are vaccinated, and half of the girls are not. They (professionals) teach health education but they don't teach it well, nobody teaches us enough."*

**Subtheme 2.2 Non-regular provision of vaccine.** Some of the professionals participated in FGD cited a shortage of supplies as a deterrent to vaccination. In particular, there is a vaccine shortage even though girls volunteer for vaccinations. Furthermore, they stated that because the campaign has a limited time frame to reach all the girls, there were opportunities lost.

IDIP-4, 17 years old unmarried Grade 9th girl participated in IDI expressed this notion as follows:

*"They give it to us when we come to school, but they don't call us from home for the vaccination. It's not far, but with all this, they don't always come in time. For example, I was vaccinated once, but for me, 6 (six) months have passed and there is still no vaccine. We also thought that we would take them when they came, but they did not come as they taught us."*

Adolescent girls mentioned irregular immunization schedules. The girls clarified that occasionally, professionals may not visit for a vaccination for two or three years.

IDIP-4, 17 years old unmarried Grade 9th girl participated in IDI expressed this notion as follows:

*"Another is that the vaccine comes and is given and then there is no continuity. Its fine if that's not the case at all. Such an interruption would appear to be another problem in itself."*

IDIP-1, 14 years old unmarried Grade 8th girl participated in IDI expressed this notion as follows:

*"I really want to get vaccinated, but the vaccine doesn't come on time. Now, the vaccine is coming after a year/two years/three years or more, and at that time, our age will go. It is very good if they come and vaccinate us every six months."*

**Subtheme 2.3 Religion prohibition.** The results demonstrated that girls' use of the HPV vaccine was discouraged by their religious activities and belief in God.

IDIP-2, 16 years old unmarried Grade 9th girl participated in IDI expressed this notion as follows:

*"There are some of my friends who run away saying that we will not take, there are those who say that religion does not allow it, because it is not allowed to take anything on Wednesdays and Fridays."*

***IDIP-3, 18 years old unmarried Grade 11th girl participated in IDI expressed this notion as follows:***

*"Some people who believe in religion will not be willing to take it because they think they don't need vaccines; they think they won't get sick in the name of Jesus. My families say, "Don't get vaccinated. We have been protected from disease from our mothers and fathers since ancient times. It was not because we were vaccinated. It is God who protected"*

**Theme 3 Parental related barriers**

**Subtheme 3.1 Lack of support from Parents.** Another challenge to HPV vaccination that some women mentioned was the absence of parental support. One participant, for instance, recalled how, as an adolescent, her mother refused to vaccinate her despite her willingness to be vaccinated and her mother's receipt of a vaccination recommendation from a healthcare provider.

IDIP-11, 18 years old unmarried not educating girl participated in IDI expressed this notion as follows:

*"My family has no knowledge, they tell me that my son "doesn't approach people who talk like that, the vaccine is something that prevents birth, it makes you sterile." They don't want me to vaccinate, just like they taught me, I run away every time the doctor comes. They don't think that I might get cervical cancer, they say, "My child, the creator will protect you, Jesus will protect us."*

IDIP-10, 15 years old unmarried Grade 8th girl participated in IDI expressed this notion as follows:

*"The family says, "Don't just go and take it, hear what it's good for and get vaccinated. Don't just take it without understanding what it is." My family says that this thing is bad, they warn us not to take it, and they don't let us. They tell us to "understand, know why it is useful, and take it".*

**Subtheme3.2 Lack of knowledge among parents.** Most parents, according to participants in both the IDI and FGD, were unaware of HPV and the vaccine. This was thought to be a major obstacle preventing their daughters from receiving vaccinations.

IDIP-4, 17 years old unmarried Grade 9thgirl participated in IDI expressed this notion as follows:

*"No one educated the family about the vaccine first. Because of that, the family does not have good knowledge about this vaccine. We also heard from teachers, but families do not have enough knowledge about vaccination. They say, "Get vaccinated if it's only at school; don't get vaccinated while on the road."*

**Sub-Theme 3.3 Low perceived risk of contracting HPV.** The parental belief that their daughters are not at risk for HPV infection was one factor associated with some girls' refusal to receive the vaccination. They believed the vaccination was unnecessary because their daughters were disciplined and not engaged in sexual activity at the time.

IDIP-3, 18 years old unmarried Grade 11thgirl participated in IDI expressed this notion as follows:

*"The perception parents say is that "vaccinations don't do anything for you. It's needed for kids who want to sleep with whoever they want, not for someone who believes in a God. It is not necessary for girls who are under the control of their parents."*

IDIP-4, 17 years old unmarried Grade 9th girl participated in IDI expressed this notion as follows:

*"Cervical cancer vaccine is good according to what they told us, but it's scary because we don't know enough, it's scary because we're children."*

## Discussion

This study aimed to assess the uptake of the HPV vaccine and associated factors among adolescent girls (14–18 years) in Merab Abaya District, Gamo Zone, and Southern Ethiopia. Factors like adolescent girl's age, schooling status, knowledge towards uptake of the HPV vaccine, attitude towards uptake of the HPV vaccine, health professional recommendation and support from their family or guardians were significant association with the uptake of HPV vaccine.

The findings of this study showed that (53.9%) (95%, CI: 49.9% − 57.9%), had ever received any dose of HPV vaccine, whereas 277 (46.1%) were not vaccinated against HPV. The findings of this study in line with those of the studies conducted in Nekemete town, eastern Ethiopia (52%) [17] and Arbaminch Town, Southern Ethiopia (50.4%) [19]. Similarly, from study in Nekemete town and Arbaminch Town could be approachable socio-demographic status, sample size, and study participants.

This finding was significantly higher than that of the study conducted in the China (11%) with [20], the Lira District of Uganda (17.6%) [21], Uganda's Kawampe (44.6%) [17], Ugandan municipality of Gulu (22%) [22], Ambo Ethiopia (44.4%) [9], and Bahir Dar City Northern Ethiopia (16%) [15]. This discrepancy might be due the fact that each area has different guidelines on HPV vaccination practices, in addition to socio-demographic differences. For instance, a study carried out in Uganda defined as the uptake of HPV vaccine obtaining all three of the recommended doses of the HPV vaccine, but in this study, those who had received the vaccine at least once were deemed to have an uptake.

However, the findings of this study were lower than those of studies conducted in Hong Kong (81.4%) [23], Brazil (83.5%) [24], Malaysia (77.9%) [25], South Africa (75%) [26], Norton, Zimbabwe (68%) [27], Menjar Shenkora, northern Ethiopia (66.5%) [10]. The discrepancy might be due to differences in the study populations, study area, study period, sample size, healthcare delivery system strategies, and vaccine accessibility. The other reason might be that the observed disparity may stem from variations in socio-demographic attributes, including educational attainment. While the majority of mothers in Brazil and South Africa who participated in the studies had at least a secondary education and above, half of the mothers in the current study did not have any formal education beyond a secondary education. In contrast to the previous research, which was conducted in institutions, the current study was conducted in the community. Furthermore, the countries where the research was conducted were developed ones, such as Brazil and South Africa, meaning that everyone there was aware of the HPV vaccine.

The first significant factor associated with the uptake of the HPV vaccine was adolescent girl's age. Older adolescent girls were 1.8 times [AOR (95% CI) 1.8 (1.14–2.9); p-value = 0.013] more likely to utilize the HPV vaccination compared to their younger adolescent girls. This is consistent with different studies conducted in Hong Kong [23], Malaysia [25], and Hawassa, Southern Ethiopia [28]. The plausible justification for these results in line might be due to the study participant similarity. But the study done in Kenya [29], Menjar district [10], and Ambo [13] shows that there is no association between age and uptake of the HPV vaccine. This difference might be due to socio-demographic and sample size differences.

In terms of variation between current schooling status, adolescent girls with current in school were 2.3 times [AOR (95% CI) 2.3 (1.22–4.4); p-value = 0.01] more likely to uptake the HPV vaccination than those who were current out of school. According to the findings, girls who attended school had a three times higher uptake of the HPV vaccination than did girls who did not. This result is in congruent with research done in Uganda [30], and Southern Ethiopia's Arbaminch City [19]. The disparity might be due to implementation of the school-based HPV vaccination delivery approach without special capacity to reach out of school girls [30,31].

This finding was supported by a qualitative study in which the majority of out-of-school adolescent girls stated that they were not receiving enough information from their healthcare provider and not getting vaccinated because they didn't know much about vaccination. *"I have never been vaccinated. My reason for not getting vaccinated is because I don't know much about vaccination."*

Knowledgeable about HPV vaccination was another significantly associated factor with uptake of the HPV vaccine. Participants who were good knowledgeable about HPV vaccination were 3.4 times [AOR (95% CI) 3.4 (2.14–5.38);

p-value = 0.000] more likely to uptake the HPV vaccination than participants who were poor knowledgeable. This is in line with different studies done in Brazil [24], Western Kenya [29], Lira District, Uganda [21], Bahir Dar [15], North Shoa [10], Debre Tabor [32], and Ambo [9]. This may be due to helps encouraging person to take HPV vaccinations and is aware of the disease's severity, mechanism of transmission, and prevention. The explanation for the possible justification between knowledge level and HPV vaccine uptake could be that as adolescent become more knowledgeable about HPV vaccine, they become more ready to receive it and also develop more positive attitudes toward it. A study conducted in Lebanon found that respondents' knowledge of the HPV vaccine was inversely associated with their uptake of the vaccine, which runs counter to our current findings [33]. The observed variation can result from variations in sample size and socio-demographic factors among study participants. This can be explained by the fact that adolescents with sufficient under-standing were given accurate information about the benefits of being vaccinated against HPV.

Most of the participants in the qualitative study explained in both IDI and FGD that most parents were unaware of HPV and the vaccine. This was thought to be a major obstacle preventing their daughters from receiving vaccinations. Also, the majority of the girls interviewed in the qualitative part knew the HPV vaccine; some revealed that they did not know the vaccine. The others complained about not receiving enough information from their healthcare provider or school, which resulted in their lack of knowledge and non-uptake of vaccines.

The other statistically significant factor with uptake of HPV vaccine was attitude. Participants who had positive attitude towards the HPV vaccine were 1.7 times [AOR (95% CI) 1.7 (1.024–2.78); p-value = 0.04] more likely to uptake the HPV vaccine than those who had a negative attitude toward it. This finding in line with study conducted in Brazil [24], Lebanon [34], Germany [35], Honghong [36], Mbale District, Uganda [37], Lira District Uganda [21], Ambo [9], Menjar District [10], Bahir Dar [15], and Arbaminch [38]. This might be due to the availability of routine vaccinations in that nation may serve as a justification, since it broadens exposure to information about the vaccine and having good perceptions of it. It also the fact that attitudes and beliefs regarding vaccine effectiveness, safety perception benefit, and convenience of access are proximal factors that significantly influences the uptake of HPV vaccination among adolescents [9].

This finding was supported by the qualitative component, which stated that adolescent girls who voiced doubts about vaccinations typically talked about how terrible the time is and how they can't trust anyone or anything during this scary period. Also, adolescent girls expressed mistrust towards vaccinations and acknowledged a widespread fear that they could render girls infertile, cause them to become members of the 666, and ultimately be used to lower the population. The other is that after receiving the injection, some girls experience dizziness, and others decide not to receive the vac-cine after learning what it is meant to do. The results demonstrated that girls' use of the HPV vaccine was discouraged by their religious activities and belief in God, and this was seen to negatively affect the uptake of the HPV vaccine.

Received advice or recommendations from health workers was another statistically significant associated factor with the uptake of the HPV vaccine. Respondents who received advice from health workers were 3.8 times [AOR (95% CI) 3.8 (2.25–6.5); p-value = 0.000] more likely to uptake the HPV vaccine than those who had not received advice from health workers. This statement in consistent with study conducted in US [39] and Lira District, Uganda [21]. It demonstrated that adolescent girls' healthcare providers' recommendations increase adolescent girls' uptake of the HPV vaccination. This might be the case when adolescent girls decide to get the HPV vaccine because they consider healthcare workers to be trustworthy sources of health information [40]. The other plausible justification showed that one particularly important pre-dictor of HPV vaccine initiation was the level of health providers' recommendations about HPV vaccination.

More than half of the participants in the qualitative part of the study stated thatnot getting specific vaccine advice from health professionals was one of the institutional barriers to receiving the HPV vaccine among adolescent girls. Also, ado-lescent girls mentioned irregular immunization schedules. The girls clarified that occasionally, professionals may not visit for a vaccination for two or three years. This reduces the uptake of the HPV vaccine.

Finally, getting support from their family or guardians was another statistically significant associated factor with the uptake of the HPV vaccine. Respondents who got support from their family or guardians to get the HPV vaccine were 7.1

times [AOR (95% CI) 7.1 (3.966–12.6); p-value = 0.000] more likely to uptake the HPV vaccine than those who did not receive support. This result is in line with a study done in the US [41], Victoria [42], and Hawassa City [28]. This may be due to school girls' vaccination age. Given how difficult it is for girls to make decisions about their health and whether to get the HPV vaccine, family and guardian support for the vaccination is crucial for interventions that let female students practice freely and have a big impact on girls' attitudes and knowledge. Hong Kong found that there is no meaningful association between the uptake of the HPV vaccine and parental or guardian support [23]. This discrepancy stems from a Hong Kong study that found parents and adolescents were equally capable of making decisions about vaccines.

Other participants in qualitative part of the study revealed the absence of parental support. One participant recalled how, as an adolescent, her mother refused to vaccinate her despite her willingness to be vaccinated. Also, the parental belief that their daughters are not at risk for HPV infection was one factor associated with some girls' refusal to receive the vaccination. They believed the vaccination was unnecessary because their daughters were disciplined and not engaged in sexual activity at the time. It is one of the barriers to the uptake of the HPV vaccine.

### Strength of the study

The strength of this study was that an explanatory sequential mixed-method study was used to support quantitative findings, which make it better insight and strengthen the study topic, and it involved new variables like health system decision making strategies and parental support for the HPV vaccine. The other is community-based nature of the study, which reflects the actual practice of the HPV vaccine among adolescent girls.

### Limitation of the study

On the contrary, the limitations of this study were that the data was subject to recall bias because the participants were questioned about when the vaccines were received, how many doses were taken, and prior HPV vaccine use. Whenever possible, the vaccination card was checked to reduce recall bias. The study in some way faced recall and self-reporting bias, since there are parameters which needs self-report. The other limitation was the cross-sectional nature of the study design; it is difficult to establish cause-and-effect relationships between maternal and child health service providers and job motivation. In next the study lacks generalizability in qualitative concern since it has main difference in sample and study area. That is applicability issue is main concern in this study since it is only done in one district, so we are unable to generalize it to other society in qualitative inquiry.

### Conclusion

Nearly fifty-four percent (54%) of the adolescent girls in Merab Abaya district, Gamo Zone, received the HPV vaccine. The identified findings of this study were that age of adolescent girls, the current schooling status of the participants, knowledge of the HPV vaccine, attitude towards the HPV vaccine, health care provider's recommendation, and family or guardians support to receive the HPV vaccine had a statistically significant association with uptake of the HPV vaccine among adolescent girls.

### Recommendation

Health facilities and schools as well as healthcare providers and parents should pay attention to factors that not uptake of the HPV vaccine among adolescent girls, there should be to raising community awareness of HPV, its vaccine, and access for adolescent girls in and out of schools, creating various training opportunities for health workers to provide adequate information on the HPV vaccine, parents need to pay for and support adolescent girls in the uptake of recommended doses of vaccine. Future researchers are recommended to conduct longitudinal studies to evaluate the cause-and-effect relationship between independent and dependent variables.

## Acknowledgments

We would like to express our gratitude to the Merab Abaya district health office included in the study and the study participants who participated in both quantitative and qualitative studies, data collectors, and supervisors for their cooperation throughout the study.

## Author contributions

**Conceptualization:** Zenebe Debena Den'o, Woldetsadik Oshine Oche.

**Data curation:** Desta Markos.

**Formal analysis:** Zenebe Debena Den'o, Tamirat Mathewos Milkano.

**Investigation:** Desta Markos.

**Methodology:** Zenebe Debena Den'o, Woldetsadik Oshine Oche, Tamirat Mathewos Milkano.

**Software:** Zenebe Debena Den'o, Tamirat Mathewos Milkano.

**Supervision:** Wondimagegn Paulos, Desta Markos.

**Validation:** Zenebe Debena Den'o, Wondimagegn Paulos, Desta Markos, Tamirat Mathewos Milkano.

**Visualization:** Desta Markos, Tamirat Mathewos Milkano.

**Writing – original draft:** Zenebe Debena Den'o, Woldetsadik Oshine Oche, Tamirat Mathewos Milkano.

**Writing – review & editing:** Wondimagegn Paulos, Woldetsadik Oshine Oche, Tamirat Mathewos Milkano.

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
