## [Decision Letter · Decision Letter 0]

24 Feb 2025

PONE-D-25-02990Human Papilloma Virus Vaccination uptake and Associated Factors among Adolescent Girls in Merab Abaya District, Gamo Zone, Southern Ethiopia: Mixed MethodsPLOS ONE

Dear Dr. Milkano,

Thank you for submitting your manuscript to PLOS ONE. After careful consideration, we feel that it has merit but does not fully meet PLOS ONE’s publication criteria as it currently stands. Therefore, we invite you to submit a revised version of the manuscript that addresses the points raised during the review process.

We look forward to receiving your revised manuscript.

Kind regards,

Morufu Olalekan Raimi, Ph.D

Academic Editor

PLOS ONE

**Journal Requirements:**

1. When submitting your revision, we need you to address these additional requirements. Please ensure that your manuscript meets PLOS ONE's style requirements, including those for file naming. The PLOS ONE style templates can be found at https://journals.plos.org/plosone/s/file?id=wjVg/PLOSOne_formatting_sample_main_body.pdf and https://journals.plos.org/plosone/s/file?id=ba62/PLOSOne_formatting_sample_title_authors_affiliations.pdf 2. We note that your Data Availability Statement is currently as follows: All relevant data are within the papers and its supporting information files. But the qualitative parts of the study, the participants were not consented to share their audio. Please confirm at this time whether or not your submission contains all raw data required to replicate the results of your study. Authors must share the “minimal data set” for their submission. PLOS defines the minimal data set to consist of the data required to replicate all study findings reported in the article, as well as related metadata and methods (https://journals.plos.org/plosone/s/data-availability#loc-minimal-data-set-definition). For example, authors should submit the following data: - The values behind the means, standard deviations and other measures reported;- The values used to build graphs;- The points extracted from images for analysis. Authors do not need to submit their entire data set if only a portion of the data was used in the reported study. If your submission does not contain these data, please either upload them as Supporting Information files or deposit them to a stable, public repository and provide us with the relevant URLs, DOIs, or accession numbers. For a list of recommended repositories, please see https://journals.plos.org/plosone/s/recommended-repositories. If there are ethical or legal restrictions on sharing a de-identified data set, please explain them in detail (e.g., data contain potentially sensitive information, data are owned by a third-party organization, etc.) and who has imposed them (e.g., an ethics committee). Please also provide contact information for a data access committee, ethics committee, or other institutional body to which data requests may be sent. If data are owned by a third party, please indicate how others may request data access. 3. One of the noted authors is a group or consortium. In addition to naming the author group, please list the individual authors and affiliations within this group in the acknowledgments section of your manuscript. Please also indicate clearly a lead author for this group along with a contact email address.

**Additional Editor Comments:**

Dear Authors,

Thank you for submitting your manuscript titled "Human Papilloma Virus Vaccination Uptake and Associated Factors among Adolescent Girls in Merab Abaya District, Gamo Zone, Southern Ethiopia: Mixed Methods" (PONE-D-25-02990). Your study addresses an important public health issue, particularly in the context of HPV vaccination among adolescent girls in Ethiopia. The mixed-methods approach is a strength, as it provides both quantitative and qualitative insights into factors influencing vaccine uptake. However, the manuscript requires major revisions to address several methodological, structural, and language-related issues before it can be considered for publication.

The reviewers have provided detailed and constructive feedback, which I encourage you to address thoroughly. Key areas for improvement include:

Language and Grammar: The manuscript contains numerous grammatical and structural errors, which hinder readability. Professional proofreading and language editing are strongly recommended.

Methodological Clarity: Please provide a clearer justification for the mixed-methods design, elaborate on sampling techniques, and ensure transparency in statistical analyses. The qualitative component, in particular, requires more detail regarding participant selection and data saturation.

Integration of Findings: The qualitative and quantitative findings are currently presented separately, leading to a lack of cohesion. A more integrated discussion would enhance the manuscript’s narrative and impact.

Ethical Considerations: While ethical approval and informed consent are mentioned, the section needs to be expanded to include details on confidentiality and parental consent for minors.

Policy and Practical Implications: The discussion and conclusion should more explicitly connect the findings to Ethiopia’s national HPV vaccination strategy and propose targeted interventions to improve uptake.

We recommend that you address these concerns thoroughly and resubmit the manuscript for further evaluation. With the suggested revisions, your study has the potential to make a meaningful contribution to the literature on HPV vaccination strategies in low-resource settings.

Thank you for considering these comments. We look forward to receiving your revised manuscript.

Reviewers' comments:

Reviewer's Responses to Questions

**Comments to the Author**

1. Is the manuscript technically sound, and do the data support the conclusions?

Reviewer #1: Partly

2. Has the statistical analysis been performed appropriately and rigorously? 

Reviewer #1: Yes

3. Have the authors made all data underlying the findings in their manuscript fully available?

Reviewer #1: Yes

4. Is the manuscript presented in an intelligible fashion and written in standard English?

Reviewer #1: Yes

5. Review Comments to the Author

**Reviewer #1:**  Key Comments & Suggested Revisions:

1. Abstract

Strengths:

• Clearly outlines the study’s background, objective, methods, results, and conclusion.

• Highlights the main statistical findings concisely.

Areas for Improvement:

• Language & Grammar: The abstract contains multiple grammatical and structural errors. Needs proofreading for better readability.

• Methodology Clarification: The mention of “two-stage sampling” and “purposive sampling” should be slightly elaborated for clarity.

• Qualitative Findings Integration: The abstract heavily focuses on quantitative results. Briefly summarizing key qualitative barriers would strengthen the abstract.

Suggestion:

• Improve clarity and grammar.

• Include qualitative findings concisely.

• Clarify key methodological aspects.

2. Introduction

Strengths:

• Provides a strong rationale for the study with a global and regional perspective on HPV vaccination.

• Discusses the burden of cervical cancer and vaccination strategies.

Areas for Improvement:

• Ethiopian Context: Needs more statistics specific to Ethiopia’s HPV vaccination program coverage.

• Study Gap & Justification: The gap in school vs. community-based adolescent girls is a key contribution but should be emphasized more clearly.

• Policy Relevance: How can findings inform Ethiopia’s national HPV vaccination strategy?

Suggestion:

• Add national data on HPV vaccination rates.

• Explicitly state how this study fills the research gap.

• Connect findings to national health policy implications.

3. Methods

Strengths:

• Provides a structured explanation of study design, population, and sampling.

• The two-stage sampling approach and use of mixed methods are appropriate.

Areas for Improvement:

• Study Design Justification: The choice of a mixed-methods design should be explicitly justified.

• Qualitative Methods Weakness:

o Lack of details on how qualitative participants were selected and how data saturation was determined.

o The qualitative component is not well-integrated with quantitative findings.

• Statistical Analysis Clarity:

o The rationale for using logistic regression should be briefly explained.

o Define statistical assumptions (e.g., multicollinearity check, goodness-of-fit test).

• Ethical Considerations:

o Ethical approval and informed consent are mentioned, but there is no discussion on how participant confidentiality was maintained.

o Parental consent for minors should be explicitly stated.

Suggestion:

• Clarify why a mixed-methods approach was chosen.

• Improve details on qualitative data collection and analysis.

• Provide more details on statistical validity checks.

• Strengthen the ethical considerations section.

4. Results

Strengths:

• Clearly presents both descriptive and inferential statistics.

• Tables and figures effectively summarize key findings.

Areas for Improvement:

• Inconsistent Data Presentation:

o The qualitative findings are summarized separately, making it difficult to integrate with the quantitative results.

o The qualitative barriers should be incorporated within the quantitative discussion to create a cohesive narrative.

• Confidence Intervals & p-values:

o Some tables lack confidence intervals for adjusted odds ratios.

o Ensure all variables in multivariate models have their respective p-values reported.

• Missing Effect Sizes for Key Findings:

o Instead of just stating associations, include an interpretation of effect sizes (e.g., "Family support increases the odds of HPV uptake by sevenfold").

Suggestion:

• Integrate qualitative and quantitative findings for a more coherent discussion.

• Ensure consistent reporting of confidence intervals and p-values.

• Interpret effect sizes meaningfully.

5. Discussion

Strengths:

• Findings are compared with other studies.

• Practical recommendations are given.

Areas for Improvement:

• Thematic Integration Weakness:

o The discussion mostly focuses on quantitative results, with minimal integration of qualitative themes.

o The qualitative findings should be used to explain why certain factors influence HPV uptake.

• Study Limitations Section is Weak:

o It should include potential biases (e.g., self-reporting bias, recall bias).

o Acknowledge limitations in the generalizability of findings.

• Policy & Practice Implications Need Strengthening:

o How can these findings inform vaccination strategies in Ethiopia?

o What targeted interventions can improve uptake?

Suggestion:

• Improve qualitative-quantitative integration.

• Expand the limitations section.

• Strengthen the practical and policy implications.

6. Conclusion

Strengths:

• Summarizes key findings concisely.

• Provides recommendations.

Areas for Improvement:

• Should explicitly connect findings to policy recommendations.

• Needs to acknowledge study limitations more clearly.

Suggestion:

• Strengthen connection to policy and public health strategies.

• Mention study limitations.

Language & Formatting Issues

Major concerns:

• Numerous grammatical and typographical errors throughout the manuscript.

• Sentence structure and wording need improvement for clarity and readability.

Suggestion:

• Professional proofreading & language editing is strongly recommended.

• Consider restructuring certain paragraphs for better flow.

Final Decision:

Major Revisions Required

Key Actions Needed Before Resubmission:

1. Improve language clarity & proofreading.

2. Enhance integration of qualitative and quantitative findings.

3. Clarify statistical methodology and reporting.

4. Strengthen ethical considerations, limitations, and policy implications.

5. Improve discussion depth and coherence.

6. PLOS authors have the option to publish the peer review history of their article (what does this mean? ). If published, this will include your full peer review and any attached files.

**Do you want your identity to be public for this peer review?** For information about this choice, including consent withdrawal, please see our Privacy Policy .

Reviewer #1: No

---

## [Author Response · Author response to Decision Letter 1]

30 May 2025

Response to academic editor and reviewersMay30, 2025

To: Morufu Olalekan Raimi, Ph.D(Academic Editor, PLOS ONE)

PONE-D-25-02990

Human Papilloma Virus Vaccination uptake and Associated Factors among Adolescent Girls in Merab Abaya District, Gamo Zone, Southern Ethiopia: Mixed Methods

Thank you for considering our manuscript and for arranging for it to be reviewed by reviewers. We have tried to address your comments and the comments / suggestions from the reviewers. We have highlighted the changes within the manuscript please find for your kind consideration. In the Response to Reviewers, we copy each of the comments or suggestions and provide the response hereunder with by number determined as Author response. We also provide a marked-up copy of the manuscript that highlights changes made to the original version and this is uploaded as a separate file labeled “Revised Manuscript with Track Changes”. Finally, we provide a cleaned version of the revised manuscript without tracked changes and this is uploaded as a separate file labeled 'Manuscript'.

We have been carefully through the peer review and have revised our paper accordingly. We feel that the paper is much improved as a result of this peer review process, and thank you for taking it to this stage. We hope we have satisfactorily addressed all the comments and hope that our paper may now be suitable for publication in your journal and we keep in touch that we are waiting yours feedback also. We hold ourselves at your entire disposition for any further information or other changes you might require.

Sincerely yours!

TamiratMathewosMilkano (Corresponding author)

Response to academic editor

1. Journal Requirements: Please ensure that your manuscript meets PLOS ONE's style requirements, including those for file naming

Author response: Thanks for reviewing our paper deeply. We reviewed our manuscript and revised it as to meet the PLOS ONE's style requirements.

2.We note that your Data Availability Statement is currently as follows: All relevant data are within the papers and its supporting information files. But the qualitative parts of the study, the participants were not consented to share their audio.

Author response: Thanks for reviewing our paper deeply. For the qualitative data consent were took from participants before data collection and after their agreement we have started collecting data, the audio agreement was also in our hand, ‘’the concept saying the participants are not consented to share their audio’’ means no consent were taken from the participants during data collection about sharing their audio for publication issue, if their audio is mandatory and needed for publication we will try to find them and share it during publication process.

Other data sets are available in time during publication and in any time.

3. One of the noted authors is a group or consortium. In addition to naming the author group.

Author response: Thanks for reviewing our paper deeply and giving constructive comments. We have reviewed and modified the authors with their roles and affiliation manuscript.

Additional Editor Comments:

1. Language and Grammar: The manuscript contains numerous grammatical and structural errors, which hinder readability. Professional proofreading and language editing are strongly recommended.

Author response: Thanks for deeply reviewing of our paper and giving recommendations to be modified and corrected. Our manuscript undergone proof reading and English grammar check by professionals and some online websites, we also acknowledge the one who helped us in proof reading of our manuscript document and we also ensure that the manuscript undergoes through English language editing and document formatting to improve its clarity and readability.

2. Methodological Clarity:Please provide a clearer justification for the mixed-methods design, elaborate on sampling techniques, and ensure transparency in statistical analyses. The qualitative component, in particular, requires more detail regarding participant selection and data saturation.

Author response: Thanks for deeply reviewing of our paper and giving recommendations to be modified and corrected.It is elaborated on both clean and track change manuscript for both.

3. Integration of Findings: The qualitative and quantitative findings are currently presented separately, leading to a lack of cohesion. A more integrated discussion would enhance the manuscript’s narrative and impact.

Author response: Thanks for deeply reviewing of our paper and we appreciate you taking the time to review our manuscript.It is elaborated on both clean and track change manuscript for qualitative and quantitative findings, by keeping cohesion and integrated discussion was done for enhancing the impact of the manuscript.

4. Ethical Considerations: While ethical approval and informed consent are mentioned, the section needs to be expanded to include details on confidentiality and parental consent for minors.

Author response: Thanks for deeply reviewing of our paper and giving recommendations to be modified and corrected. In any situation of research work the ethical consideration were appropriately applied, considering minors in time of research work was also applied by making appropriate communication with their parents. Verbal consent wasundertaking for the issue. Major ethical principles like beneficence, non-maleficence, autonomy, and justice were followed in any situation of research work.

5. Policy and Practical Implications: The discussion and conclusion should more explicitly connect the findings to Ethiopia’s national HPV vaccination strategy and propose targeted interventions to improve uptake.

Author response: Thanks for deeply reviewing of our paper and we appreciate you taking the time to review our manuscript. We have considered all the policy and strategies currently working in Ethiopia for HPV vaccination implementation and indication. All the discussion conclusion with recommendations were done based on current policy of Ethiopia.

Reviewers'commentsto the Author

1. Is the manuscript technically sound, and do the data support the conclusions?The manuscript must describe a technically sound piece of scientific research with data that supports the conclusions. Experiments must have been conducted rigorously, with appropriate controls, replication, and sample sizes. The conclusions must be drawn appropriately based on the data presented.

Reviewer #1: Partly:Author response: Thank you. We appreciate you taking the time to review our manuscript. We have considered all your implication and tried to make our manuscript technically sound and rigorous work conducted in any step of the work for publication and for valid scientific finding, appropriate and relevant conclusion was drawn from the research work.

2. Has the statistical analysis been performed appropriately and rigorously?

Reviewer #1: Yes

Author response: Thank you. We appreciate you taking the time to review our manuscript

3. Have the authors made all data underlying the findings in their manuscript fully available?

The PLOS Data policy requires authors to make all data underlying the findings described in their manuscript fully available without restriction, with rare exception (please refer to the Data Availability Statement in the manuscript PDF file). The data should be provided as part of the manuscript or its supporting information, or deposited to a public repository. For example, in addition to summary statistics, the data points behind means, medians and variance measures should be available. If there are restrictions on publicly sharing data. e.g., participant privacy or use of data from a third party—those must be specified.

Reviewer #1: Yes

Author response: Thank you. We appreciate you taking the time to review our manuscript

4. Is the manuscript presented in an intelligible fashion and written in standard English?

Reviewer #1: Yes

Author response: Thank you. We appreciate you taking the time to review our manuscript. We have deeply edited the writing to make it in intelligible fashion and English language editing was intensively done by software’s and website.

5. Review Comments to the Author

Author response: Thank you. We appreciate you taking the time to review our manuscript. There is no dual publication were done. Ethical issues were fully considered by using basic research ethics principles.

Reviewer #1: Key Comments & Suggested Revisions:

1. Abstract

Strengths: Clearly outlines the expected contents

Author response: Thank you. We appreciate you taking the time to review our manuscript.

Areas for Improvement: Improve clarity and grammar, include qualitative findings concisely, Clarify key methodological aspects

Author response: Thank you. We appreciate you taking the time to review our manuscript. All the comments were modified in both manuscript clean and with track change

2. Introduction

Author response: We appreciate the reviewer for taking the time to review our manuscript and for giving the constructive comments and suggestions as you provided as strength.

Areas for Improvement and suggestions: Improve on more statistic data on Ethiopian context, appropriate justification on gaps, support policy relevance, add national data on HPV vaccination rates and connection of finding to national health policy.

Author response: Thank you. We appreciate you taking the time to review our manuscript. All the comments were modified in both manuscript clean and with track change.

Authors tried to incorporate and correct all the suggestions and areas needed to be improved. In the content suggesting that add more statistical data from Ethiopian context: there are less amount of study done which are relevant for our study were included. All the needed and relevant justifications describing gaps wereincorporated in the manuscript in both clean and with track change. Justification supporting the policy relevance and connecting finding to the national health policy were also modified in manuscript in both clean and track changed.

3. Methods

Author response: We appreciate the reviewer for taking the time to review our manuscript and for giving the constructive comments and suggestions as you provided as strength.

Areas for Improvement and suggestions: Improve on more on area of study design in both quantitative and qualitative part, ethical consideration, validity checking mechanism

Author response: Thank you. We appreciate you taking the time to review our manuscript. All the comments were modified in both manuscript clean and with track change.

Authors tried and corrected all comments and suggestions to be improved in methodology part.

Study Design Justification: The reason for using mixed method in the study were there were findings that support the uptake of HPV vaccine were affected by cultural issues and other factors which needs in-depth interview and supporting the quantitative data.

Qualitative Methods Weakness:Integration of qualitative with quantitative is done in manuscript. The data saturation is happened at participant 8 from minimum expected sample size to be needed which is 16.

Statistical Analysis Clarity:The reason for using logistic regression analysis for analysis is most of the studies related to the title are done in logistic analysis and the concept mainly done by making it dichotomous.

Multicolinearity test is method checking two or more independent variables are related with each other and can bring effect on outcome variable. The test was done by checking variance inflation factor and tolerance teste in SPSS. The GoF or goodness of fit test is done by Hosmer lemshow test and the variables in significant range or <0.05 took as good model fit.

Ethical Considerations: all type of ethical principles were considered and conducted in different steps of research process, regarding minors’ discussion between their parents and if they were in school their respective school director was conducted regarding the research process, maintaining confidentiality were taken in any phase of research which was also done by common agreement among participants and researchers.

Suggestions:Improve details on qualitative data collection and analysis:The interview was conducted by the principal investigator.A semi-structured interview guide was prepared and used in both the in-depth interview and focus group discussion. The principal investigator wastaken video-based lectures to upgrade his capacity on how to approach participants, how to handle them during interviews, how to probe, and how to ask sensitive questions. FGDs’ participants were appointed one week before the meeting for discussion, and participants consisting of(health professionals, and teachers).

The average time used for IDI and FGD were 24:30 and 1:11:35 minutes respectively. Study participants were informed about the purpose of the study and what the research is all about before recruiting them to the study at first contact. Then the interview at quiet places with maximum privacy had been chosen for both IDIs and FGDs. On the other hand, issues of safety for both the principal investigator and the study participants was the main considerations while selecting that place. The topic guide addressed all issues related to barriers with human Papilloma virus uptake and other issues relevant to the objectives of the study. The principal investigator had conductedthe discussions, while one assistance co-facilitator had assigned during the FGDsto manage discussion time and facilitate recording. IDI and FGD were guided by an experienced person fluent in the local language (Gammotho) and English with the researcher.The interview had been recorded on tapes, which were later transcribed. Detailed field notes had been taken.

Provide more details on statistical validity checks.Statistical validity of the tool initially was done by expert and additional validity test was also done. To maintain the internal consistency of the instrument, the items' Cronbach's alpha coefficient was calculated. For qualitative data, one research assistance having qualitative data collection experience and principal investigators using interview guides (open-ended) was used. The data normality was done by Shapiro wilk test.

4. Results:

Author response: We appreciate the reviewer for taking the time to review our manuscript and for giving the constructive comments and suggestions as you provided as strength.

Areas for Improvement and suggestions: Improve on more on area of data Presentation, CI and P-values, missing effect sizes for key findings and its interpretation.

Author response: Thank you. We appreciate you taking the time to review our manuscript. All the comments were modified in both manuscript clean and with track change.

Authors tried and corrected all comments and suggestions to be improved in result part.

Inconsistent Data Presentation:The presentation of qualitative part in different were, since it is reporting and the result reporting needed to be done by itself. The discussion part was done by merging both qualitative and quantitative together, analysis generally done by quantitative followed by qualitative approach, integration of both quantitative and qualitative findings for a more coherent discussion and all needed modifications were done in both manuscripts.

Confidence Intervals & p-values:variables in tables which lack confidence interval were modified in manuscript both in clean and with track changes. The reason for not putting P- value is mainly due to that they are not significantly associated. Writing P-vale and confidence interval with effect size were mainly done factor which affects the uptake of H

---

## [Editor Report · Decision Letter 1]

23 Jun 2025

PONE-D-25-02990R1Human Papilloma Virus Vaccination uptake and Associated Factors among Adolescent Girls in Merab Abaya District, Gamo Zone, Southern Ethiopia: Mixed MethodsPLOS ONE

Dear Dr. Milkano,

Thank you for submitting your manuscript to PLOS ONE. After careful consideration, we feel that it has merit but does not fully meet PLOS ONE’s publication criteria as it currently stands. Therefore, we invite you to submit a revised version of the manuscript that addresses the points raised during the review process.

We look forward to receiving your revised manuscript.

Kind regards,

Morufu Olalekan Raimi, Ph.D

Academic Editor

PLOS ONE

Journal Requirements:

Additional Editor Comments:

After reviewing the manuscript and the authors' responses to the reviewers' comments, here is my assessment and recommendation:

Strengths of the Manuscript:

1. The mixed-methods approach (quantitative and qualitative) provides a robust analysis of HPV vaccine uptake and associated factors, enhancing the depth of understanding.

2. The study aims to assess HPV vaccine uptake among adolescent girls in a specific region, addressing a gap in existing literature.

3. The sampling technique, data collection, and analysis are well-described, ensuring reproducibility.

4. The study identifies key factors influencing vaccine uptake (knowledge, attitude, health worker recommendations, family support) and barriers (mistrust, irregular provision, lack of information).

5. Ethical approval and consent procedures are documented, and confidentiality measures are highlighted.

Areas Addressed in Revisions:

• The authors state that the manuscript underwent professional proofreading and language editing.

• Additional details on sampling techniques, qualitative data saturation, and statistical validity checks were provided.

• The discussion now better integrates qualitative and quantitative results.

• Expanded details on confidentiality and parental consent for minors were included.

• The discussion now explicitly connects findings to Ethiopia’s national HPV vaccination strategy.

Remaining Concerns:

1. The qualitative audio data are not shared due to consent issues, which may limit transparency. However, the authors offer to provide it if mandatory.

2. The study’s focus on one district may limit broader applicability, though this is acknowledged as a limitation.

3. While proofreading was done, some minor grammatical or structural errors may persist.

Recommendation:

Accept the Manuscript with Minor Revisions

The manuscript is well-structured, addresses an important public health issue, and has incorporated most reviewer feedback. The remaining concerns are minor and do not detract from the study's overall quality and contributions.

Justification for Acceptance:

• The study provides valuable insights into HPV vaccine uptake in a low-resource setting, with practical recommendations for improving vaccination rates.

• The mixed-methods approach strengthens the findings.

• Ethical and methodological rigor is demonstrated.

• The revisions have adequately addressed the reviewers' comments.

Proceed with acceptance, pending these minor checks.

Final Decision: Accept with Minor Revisions

---

## [Author Response · Author response to Decision Letter 2]

27 Jul 2025

Response to academic editor and reviewers July 28, 2025

To: Morufu Olalekan Raimi, Ph.D(Academic Editor, PLOS ONE)

PONE-D-25-02990

Human Papilloma Virus Vaccination uptake and Associated Factors among Adolescent Girls in Merab Abaya District, Gamo Zone, Southern Ethiopia: Mixed Methods

Thank you for considering our manuscript and for arranging it to be reviewed by reviewers. We have tried to address your comments and there is no attachment describing about reviewers point in corresponding author email and website. We have highlighted the changes within the manuscript by considering comments. The response to comments were hereunder determined as Author response. We also provide a marked-up copy of the manuscript that highlights changes made to the original version and this is uploaded as a separate file labeled “Revised Manuscript with Track Changes”. Finally, we provide a cleaned version of the revised manuscript without track changes and this is uploaded as a separate file labeled 'Manuscript'.

We have been carefully through the peer review and have revised our paper accordingly. We feel that the paper is much improved as a result of this peer review process, and thank you for taking it to this stage. We hope we have satisfactorily addressed all the comments and hope that our paper may now be suitable for publication in your journal and we keep in touch that we are waiting yours feedback also. We hold ourselves at your entire disposition for any further information or other changes you might require.

Sincerely yours!

Tamirat Mathewos Milkano (Corresponding author)

Response to academic editor

1. Journal Requirements: Please review your reference list to ensure that it is complete and correct. If you have cited papers that have been retracted, please include the rationale for doing so in the manuscript text, or remove these references and replace them with relevant current references. Any changes to the reference list should be mentioned in the rebuttal letter that accompanies your revised manuscript. If you need to cite a retracted article, indicate the article’s retracted status in the References list and also include a citation and full reference for the retraction notice.

Author response:Thanks for reviewing our research work deeply. All the references were reviewed and no change have been made, retracted paper was not used in our work.

Additional Editor Comments:

After reviewing the manuscript and the authors' responses to the reviewers' comments, here is my assessment and recommendation:

Strengths of the Manuscript:

1. The mixed-methods approach (quantitative and qualitative) provides a robust analysis of HPV vaccine uptake and associated factors, enhancing the depth of understanding.

Author response: Thanks for reviewing our research work deeply. The authors want to appreciate the editor for giving constructive comments and suggestions on the area of relevance of the design and its implication on selected title.

2. The study aims to assess HPV vaccine uptake among adolescent girls in a specific region, addressing a gap in existing literature.

Author response: Thanks for reviewing our research work deeply andgiving constructive comments and suggestions.

3.The sampling technique, data collection, and analysis are well-described, ensuring reproducibility.

Author response:Thanks,from bottom of our heart for reviewing our research work deeply andgiving constructive comments and suggestions.

4. The study identifies key factors influencing vaccine uptake (knowledge, attitude, health worker recommendations, family support) and barriers (mistrust, irregular provision, lack of information).

Author response: Thanks, from bottom of our heart for reviewing our research work deeply andgiving constructive comments and suggestions.

5. Ethical approval and consent procedures are documented, and confidentiality measures are highlighted.

Author response: Thanks, from bottom of our heart for reviewing our paper deeply andgiving constructive comments and suggestions.

Areas Addressed in Revisions:

• The authors state that the manuscript underwent professional proofreading and language editing.

• Additional details on sampling techniques, qualitative data saturation, and statistical validity checks were provided.

• The discussion now better integrates qualitative and quantitative results.

• Expanded details on confidentiality and parental consent for minors were included.

• The discussion now explicitly connects findings to Ethiopia’s national HPV vaccination strategy.

Author response: Thanks, from bottom of our heart for reviewing our paper deeply andgiving constructive comments and suggestions. The authors also thank again the editor for clearly reverifying what we have addressed the correction of first review from reviewers and editor.

Remaining Concerns:

1. The qualitative audio data are not shared due to consent issues, which may limit transparency. However, the authors offer to provide it if mandatory.

Author response: Thanks, from bottom of our heart for reviewing our paper deeply andgiving constructive comments and suggestions. The authors ensure that we will share the audio data in any sharing method if it is mandatory.

2. The study’s focus on one district may limit broader applicability, though this is

acknowledged as a limitation.

Author response: Thanks, from bottom of our heart for reviewing our paper deeply andgiving constructive comments and suggestions. The limitation of the study mainly applicability part is described in the manuscript in clean and track changed type.

3. While proofreading was done, some minor grammatical or structural errors may persist.

Author response: Thanks, from bottom of our heart for reviewing our paper deeply andgiving constructive comments and suggestions. We ensure that we are also working on through the time of publication and we have also done some editing in manuscript also.

Recommendation:

Accept the Manuscript with Minor Revisions

The manuscript is well-structured, addresses an important public health issue, and has incorporated most reviewer feedback. The remaining concerns are minor and do not detract from the study's overall quality and contributions.

Author response: Thanks, from bottom of our heart for reviewing our paper deeply andgiving constructive comments and suggestions. We will keep in touch for any feedback regarding our research we ensure that we work on it anything needed, thanks again.

Justification for Acceptance:

• The study provides valuable insights into HPV vaccine uptake in a low-resource setting, with practical recommendations for improving vaccination rates.

• The mixed-methods approach strengthens the findings.

• Ethical and methodological rigor is demonstrated.

• The revisions have adequately addressed the reviewers' comments.

• Proceed with acceptance, pending these minor checks.

Author response: Thanks, from bottom of our heart for reviewing our paper deeply and giving constructive comments and suggestions. We will keep in touch for any feedback regarding our research we ensure that we work on it anything needed, thanks again.

Final Decision:

Accept with Minor Revisions

---

## [Editor Report · Decision Letter 2]

6 Aug 2025

Human Papilloma Virus Vaccination uptake and Associated Factors among Adolescent Girls in Merab Abaya District, Gamo Zone, Southern Ethiopia: Mixed Methods

PONE-D-25-02990R2

Dear Authors Dr.

We’re pleased to inform you that your manuscript has been judged scientifically suitable for publication and will be formally accepted for publication once it meets all outstanding technical requirements.

Kind regards,

Morufu Olalekan Raimi, Ph.D

Academic Editor

PLOS ONE

Additional Editor Comments (optional):

After all necessary corrections has been made. This manuscript meets the high standards of scientific rigor, original contribution, and public health relevance expected by PLOS ONE. Its mixed-methods approach enriches understanding of HPV vaccine uptake barriers in a critical region, providing evidence to guide effective interventions.

Therefore, the manuscript should be accepted for publication to advance global efforts in cervical cancer prevention and adolescent health promotion.

Kind regards

Prof. Morufu Olalekan Raimi
---

## [Editor Report · Acceptance letter]

PONE-D-25-02990R2

PLOS ONE

Dear Dr. Milkano,

I'm pleased to inform you that your manuscript has been deemed suitable for publication in PLOS ONE. Congratulations! Your manuscript is now being handed over to our production team.

Kind regards,

on behalf of

Prof Morufu Olalekan Raimi

Academic Editor

PLOS ONE